behaviour/ecology/health and disease and epidemiology

Chiroptera, consistent individual differences, host–pathogen dynamics, super-spreader

**Author for correspondence:**
Quinn M. R. Webber
e-mail: webber.quinn@gmail.com

†Present address: Cognitive and Behavioural Ecology Interdisciplinary Program, Memorial University of Newfoundland, St John's, NL, Canada.

# Personality affects dynamics of an experimental pathogen in little brown bats

## Quinn M. R. Webber† and Craig K. R. Willis

Department of Biology and Centre for Forest Interdisciplinary Research (C-FIR), University of Winnipeg, Winnipeg, MB, Canada

  QMRW, 0000-0002-0434-9360

Host behaviour can affect host–pathogen dynamics and theory predicts that certain individuals disproportionately infect conspecifics during an epidemic. Consistent individual differences in behaviour, or personality, could influence this variation with the most exploratory or sociable individuals most likely to spread pathogens. We quantified exploration and sociability in little brown bats (*Myotis lucifugus*) and then experimentally manipulated exposure to a proxy pathogen (i.e. ultraviolet (UV) fluorescent powder) to test two related hypotheses: (i) more sociable and more exploratory individuals would be more likely to transmit infections to other individuals, and (ii) more sociable and more exploratory individuals uninfected with an invading pathogen would be more likely to acquire infections. We captured 10 groups of 16 bats at a time and held each group in an outdoor flight tent equipped with roosting-boxes. We used hole-board and Y-maze tests to quantify exploration and sociability of each bat and randomly selected one individual from each group for 'infection' with non-toxic, UV fluorescent powder. Each group of 10 bats was released into the flight tent for 24 h, which represented an experimental infection trial. After 24 h, we removed bats from the trial, photographed each individual under UV light and quantified infection intensity from digital photographs. As predicted, the exploratory behaviour of the experimentally infected individual was positively correlated with infection intensity in their group-mates, while more exploratory females had higher pathogen acquisition. Our results highlight the potential influence of host personality and sex on pathogen dynamics in wildlife populations.

## 1. Introduction

The recent emergence of high-profile infectious diseases of wildlife has caused concern for human public health [1] and wildlife conservation [2]. Most infectious diseases of humans are zoonotic (i.e. passed from animals to humans) and originate in wildlife [3].

At the same time, wildlife diseases with conservation implications are also increasing and causing devastating impacts on host species [2,3]. Multiple human impacts can affect emergence and persistence of wildlife infectious diseases including habitat alteration, agriculture and human population expansion [4]. In addition to external factors, characteristics of hosts can influence host–pathogen dynamics and the emergence and persistence of wildlife disease.

Host behaviour represents one category of traits that can influence pathogen dynamics [5,6]. The role of host behaviour can be considered in terms of both pathogen transmission (i.e. when infected individuals spread a pathogen to susceptible hosts) and acquisition (i.e. when susceptible hosts interact with infected hosts or substrates and acquire infections) [7]. Consistent individual differences in behaviour can influence patterns of association with implications for pathogen dynamics. For example, some individuals interact with more conspecifics than others and these individuals should be more likely to spread and acquire pathogens [8,9]. The phenomenon has been termed 'super-spreading', and is the disproportionate role some individuals have in pathogen transmission [8]. Super-spreading can be influenced by host physiology and immunology but the role of host behaviour has received less attention despite its potential importance [9].

Consistent individual differences in behaviour, or animal personality [10], could represent a behavioural phenotype affecting pathogen transmission and acquisition [11]. Some personality traits are associated with an increased risk of acquiring parasites from conspecifics [12] or environmental reservoirs [13]. For instance, sociability, defined as an individual's reaction to the presence or absence of conspecifics, could be particularly important [14] because sociable individuals with high contact rates should have more opportunities to spread infection [12]. However, despite theoretical models [11,15] and some observational data [13,16,17], experimental tests of this hypothesis are generally lacking.

Bats are an important taxon to understand in terms of their host–pathogen interactions. Bats are known hosts of pathogens with public health significance (e.g. rabies virus, Hendra virus) and suspected as hosts of the ancestors of emerging coronaviruses like SARS-CoV-2 [18–20]. In addition, some hibernating bat species in North America are imperilled by white-nose syndrome (WNS), a disease with devastating conservation impacts [21,22]. Many bat species are highly gregarious and females of many temperate species are colonial, forming maternity colonies during spring and early summer while males roost solitarily or in small groups [23]. During late summer and autumn temperate bats engage in 'swarming', during which they fly in and around potential hibernacula and mate promiscuously before hibernation [24]. During swarming, social associations tend to be ephemeral and non-preferential [25]. Thus, although temperate bats sexually segregate in spring and early summer, contact between males and females is probably limited to mating, and, beyond mating, it remains unknown the degree to which males and females interact socially during swarming. The role of sex-specific social and roosting interactions could, therefore, influence the relationship between personality and pathogen transmission dynamics. There is evidence of a relationship between sex-specific personality and parasite dynamics during autumn swarming in bats, but it is limited to a single observational study which found a correlation between personality and both ectoparasite prevalence and intensity for female, but not male, little brown bats, *Myotis lucifugus* [16].

WNS is an emerging infectious disease caused by the fungal pathogen *Pseudogymnoascus destructans*, and has resulted in recent mass mortality of *M. lucifugus* and other bat species [22,26]. Transmission of *P. destructans* appears to be related to behavioural variation among both individuals and species [27]. One hypothesis explaining the rapid expansion of *P. destructans* is that a small proportion of bats that visit multiple hibernacula during swarming may be responsible for the majority of transmission events. Although transmission is assumed to occur during hibernation, some proportion of *P. destructans* transmission must occur during swarming. In addition to *P. destructans*, transmission of viral pathogens [28] as well as ectoparasites [29] are presumably socially transmitted during swarming. Swarming aggregations, therefore, represent an opportunity to assess the role of individual behaviour as a mediating factor in the transmission of parasites and pathogens.

We captured adult little brown bats during swarming and housed them in a semi-natural flight enclosure to experimentally manipulate dynamics of a proxy for a contagious pathogen (i.e. ultraviolent (UV) fluorescence powder, [27]) and the behavioural composition of groups of bats to test two hypotheses. First, we tested whether behavioural type of infected hosts would influence pathogen transmission to uninfected conspecifics within groups of bats. We predicted that more sociable, explorative and active individuals would spread the pathogen more effectively resulting in higher infection intensities for other bats. Second, we tested whether an uninfected individual's behavioural type would affect its level of pathogen acquisition. We predicted that more sociable, explorative and active individuals would acquire higher infection intensities because these individuals would be most

likely to interact with the originally infected conspecific, other secondarily infected conspecifics, as well as the environment. For both hypotheses, we partitioned the role of sex to determine its effect as a potential moderator of pathogen transmission and acquisition and we quantified the magnitude of all effects to assist with the design of future studies.

# 2. Material and methods

## 2.1. Study sites and subjects

From 29 July to 14 September 2014, we captured swarming little brown bats outside the entrance of St George Bat Cave (approx. 50 km north of Fisher River First Nation, 51°44′ N, 97°36′ W), using a harp trap and then transported them approximately 15 km to a field station for processing. Juvenile bats were identified based on degree of ossification of the fifth metacarpal–phalangeal joint [30] and released immediately. During transport bats were held in disposable paper bags inside a ventilated picnic cooler to dampen noise. Once at the field station we recorded body mass (±0.1 g) and implanted a uniquely coded passive transponder (PIT tag, Trovan Ltd ID 100-01, Douglas, UK) subcutaneously between the scapulae for permanent identification of each individual.

## 2.2. Experimental design and housing

We conducted 10 trials, each of which involved holding 16 different adult bats in captivity for approximately 48 h (electronic supplementary material, table S1). The sex ratio of groups depended on the bats captured on a given night, but mean sex ratio of the 10 groups was approximately equal (48.6 ± 6.6% female; electronic supplementary material, table S1). Bats were housed in a nylon mesh flight tent (2.75 × 2.75 × 2.75 m) with a shaded roof but otherwise open to ambient conditions. Captive bats were provided ad libitum water and mealworms (larval *Tenebrio molitor* gut-loaded with Herptivite beta carotene multivitamins and Repashy nutrient supplements). The flight tent was outfitted with four single-chamber bat boxes (volume = 3000 cm$^3$) constructed from cleanable 'vinyl plywood' and mounted on 1.5 m stands. The flight tent and boxes were cleaned using alkyl-based disinfectant wipes (Lysol, Reckitt Benckiser, Mississauga, Canada) between trials to prevent potential contamination by *P. destructans* and/or residual scent between trials. Ambient temperature ($T_a$) can influence roost selection decisions of bats [31,32]. We, therefore, deployed a temperature data logger (HOBO Micro Station—H21-002, Onset Computer Corporation, Cape Cod, Massachusetts, USA) adjacent to the flight tent to continuously record $T_a$ at 15 min intervals.

## 2.3. Ultraviolet powder 'infection'

We used UV fluorescent powder (Signal Green Pigment, DayGlo Color Corp., Cleveland, Ohio, USA) as a proxy for an infectious pathogen in bats. UV powder has been successfully used to understand avenues of pathogen transmission in bats (e.g. *P. destructans*, many viruses or parasites, [27]). Importantly, UV powder does not engage host immune defences and elicit a sickness behaviour response, as occurs for little brown bats during advanced infection with *P. destructans* [33] and vampire bats (*Desmodus rotundus*) immune-challenged with a lipopolysaccharide [34,35]. However, as shown by Hoyt *et al.* [27], UV powder reliably mimics infection dynamics of a contact pathogen like *P. destructans*. This may be especially true on the timescale of our experiment trials, which were much shorter than the time that would be required for slow-growing *P. destructans* to establish an infection and cause behavioural changes in bats. Using UV powder, therefore, enabled us to investigate the effects of behaviour on the short-term dynamics of a pathogen like *P. destructans* as it infects a new group of bats. In addition, while immune challenge with lipopolysaccharide mimics bacterial infections and alters host behaviour in at least one bat species [34,35], UV powder could reliably approximate transmission dynamics of bat viral infections because bats appear to inhibit or mitigate the immune response to many viruses that would typically cause sickness behaviour (and fever) in other mammals [36,37].

On the second night of captivity for each trial (approx. 24 h after capture), we randomly selected a single bat from the flight tent to 'infect' with UV fluorescent powder. We standardized infection 'dose' by evenly covering the ventral and dorsal surfaces of each wing membrane. Within 5 min of infection this bat was released back into the flight tent and allowed to associate with the rest of the group for the next 24 h. At the end of each trial (i.e. approx. 48 h after capture), we removed all bats

from the flight tent and isolated each individual in disposable paper bags. We gently restrained each bat on a flat table and photographed them in the dark under UV light (Ledwholesalers LED flashlights, 395 nm, Blacklight) using a digital camera (Digital Rebel XTi, Canon Inc., Japan) to quantify UV powder infection. A scaling item (Canadian dime, radius = 18 mm) was placed in the field of view of each photograph and the camera was mounted on a tripod to ensure images were consistent. We took six photographs of each individual: ventral and dorsal surfaces of the right and left wings, and ventral and dorsal torso (electronic supplementary material, figure S1).

We used ImageJ (v. 1.47v, National Institutes of Health, USA) to quantify infection intensity of each individual as the proportion of the total body surface infected [38,39]. The entire body surface visible in the photo (i.e. wing or torso) was measured by outlining the visible area in each wing or torso photograph. Occasionally the handler occluded a small part of the wing or torso so an estimate of this area was included in the total. We then measured the area of the body surface covered with visible UV powder and calculated the proportion of total body surface that reflected UV light for each bat.

## 2.4. Measuring personality

We quantified personality traits for each individual using hole-board and Y-maze tests [40–42]. Traits measured for little brown bats in both tests are repeatable within individuals across time and are correlated with ecologically relevant roosting and social behaviours [42]. Although we did not measure repeatability in this study, all personality traits measured here have previously been shown to be weakly to highly repeatable (i.e. activity hole-board: $r = 0.17$; exploration: $r = 0.27$; activity Y-maze: $r = 0.41$; sociability: $r = 0.13$) over an 11–14 day period [42]. For bats, the hole-board test quantifies activity (i.e. general movement patterns) and exploration (i.e. reaction to a novel object or situation). The hole-board consisted of a test chamber (57 cm wide by 42 cm tall by 14 cm deep) with a transparent plexiglass cover, and window screening on the back surface to facilitate climbing. Four blind holes (2 cm wide by 1 cm deep) were positioned in the backboard. A start chamber (16 by 8 cm diameter tube) was fastened to the base of the test, with a sliding door to separate the animal from the main chamber. The test apparatus was hung vertically so bats could crawl on the backboard and explore the blind holes. At the start of each trial, a bat was placed in the start chamber and the sliding door was opened. Bats were given 60 s to voluntarily enter the test before being gently pushed in using a plastic plunger.

For bats, the Y-maze test assesses individual sociability (i.e. reaction to the presence or absence of a conspecific). Our test arena consisted of a Y-shaped chamber of plexiglass (long end: 37 cm long; forked ends: 20 cm long, 6 cm wide by 10 cm tall). For each experimental trial, we randomly selected one bat from the capture site, in addition to the 16 experimental individuals, as the designated 'stimulus' bat. Each trial began when we placed an experimental bat at the long end of the Y-maze and concluded after 5 min. The stimulus bat was held in a stainless steel mesh cage (20 × 20 × 20 cm) located at the end of one of the arms of the Y and its position was randomly selected for each trial. An identical empty cage was placed at the end of the other arm of the Y. We ensured that focal individuals were isolated from the stimulus bat for at least 48 h prior to testing [43]. An important caveat associated with the Y-maze test is the inability to account for variation in the behaviour of the stimulus bat [42].

Hole-board and Y-maze trials occurred overnight and were video-recorded in the dark under infrared illumination (Sony AVCHD NightShot handycam HDR-XR550). After completion of both hole-board and Y-maze tests, bats were immediately returned to the flight tent and the testing apparatus was cleaned with disinfecting wipes (Lysol, Reckitt Benckiser, Mississauga, Canada) and allowed to air dry for 10 min to prevent residual scent from influencing the behaviour of subsequent bats. To maximize efficiency in the field we used two identical test chambers (both hole-board and Y-maze tests) interchangeably so that one could dry while the other was in use. We quantified behavioural variables for both hole-board and Y-maze tests, that have been used to quantify personality in rodents [44] and bats [40,42] (table 1 for detailed descriptions).

## 2.5. Statistical analysis

All statistical analyses were conducted in R [45]. We used two series of principal component analyses (PCA, function 'prcomp') to reduce the large number of behavioural variables into components reflective of personality traits. Prior to conducting each PCA, we confirmed that correlations existed among behavioural variables using Bartlett's test and confirmed sampling adequacy using the Kaiser–Meyer–Olkin (KMO) test. We scaled and centred raw data by subtracting variable mean values from

**Table 1.** Ethogram of 12 behaviours quantified in video recordings of hole-board and Y-maze tests conducted on 160 little brown bats. Behaviours reflect personality dimensions quantified in bats by Menzies *et al*. [40], Kilgour *et al*. [41] and Webber and Willis [42].

| behaviour | description | test |
| --- | --- | --- |
| locomotion | total duration of time a bat spent moving; either crawling or climbing | hole-board and Y-maze |
| echolocation | total duration of time spent stationary, presumably echolocating | hole-board and Y-maze |
| grooming | total duration of time a bat spent grooming; either chewing or scratching | hole-board and Y-maze |
| line crossing | intersecting vertical and horizontal grid lines separate the hole-board test into four quadrants; number of times a bat crossed a grid line | hole-board |
| flight attempts | number of flight attempts | hole-board |
| frequency of head dips | total number of times an individual investigated blind holes on the backboard | hole-board |
| latency to head dip | time from beginning of trial a bat first investigated one of the blind holes on the backboard | hole-board |
| latency to enter | time from entry into the start chamber a bat entered the test arena | hole-board |
| line crossing | number of times a bat exited an arm and subsequently entered a different arm of the Y-maze test | Y-maze |
| relative time spent within 10 cm of the stimulus bat | total duration of time the focal bat spent inside 10 cm of the end of the stimulus arm divided by the total duration of each test (300 s) | Y-maze |
| latency to social | time from beginning of trial until the focal bat first entered within 10 cm of the stimulus bat | Y-maze |

each individual value and dividing by the variable standard deviation using the 'prcomp' function in R. This generates a dataset with mean values of zero, which ensures that the first component describes the most variance. We retained components based on the Kaiser–Guttman criterion, i.e. eigenvalues > 1 [46], and the parallel analysis method [47]. Given recent criticism of the Kaiser criterion [47], in cases where the Kaiser criterion and parallel analysis disagreed, we visually inspected PCA results using a scree plot and chose the number of components based on the most conservative result. For the hole-board PCA we condensed behavioural variables to determine activity and exploration scores for each individual [40]. For the Y-maze PCA we condensed behavioural variables to determine activity and sociability [41] scores for each individual.

Prior to analysis, we tested for normality and collinearity among variables using variance inflation factors (all VIFs < 2.2, indicating low multicollinearity among variables). To test our hypotheses, we conducted two series of analyses. First, we used a single linear model to assess the potential for the personality of an infected host to influence transmission from that host to uninfected individuals (hypothesis 1: infection transmission). For this analysis we treated each trial as an experimental unit and assessed how behavioural traits of the originally infected individual influenced the mean infection intensity of bats in each trial. We used log-transformed values of mean UV powder infection intensity for uninfected bats as the dependent variable and sex, $PC1_H$ (activity), $PC2_H$ (exploration), $PC1_Y$ (activity) and $PC2_Y$ (sociability) of the infected host, along with $T_a$ at dawn ($T_{a\text{-dawn}}$) as predictor variables.

Second, we used a single linear mixed model [48] to assess the role of candidate variables on the risk of acquiring infection (hypothesis 2: infection acquisition). We used each individual's log-transformed UV powder infection intensity as the dependent variable and included sex, $PC1_H$ (activity), $PC2_H$

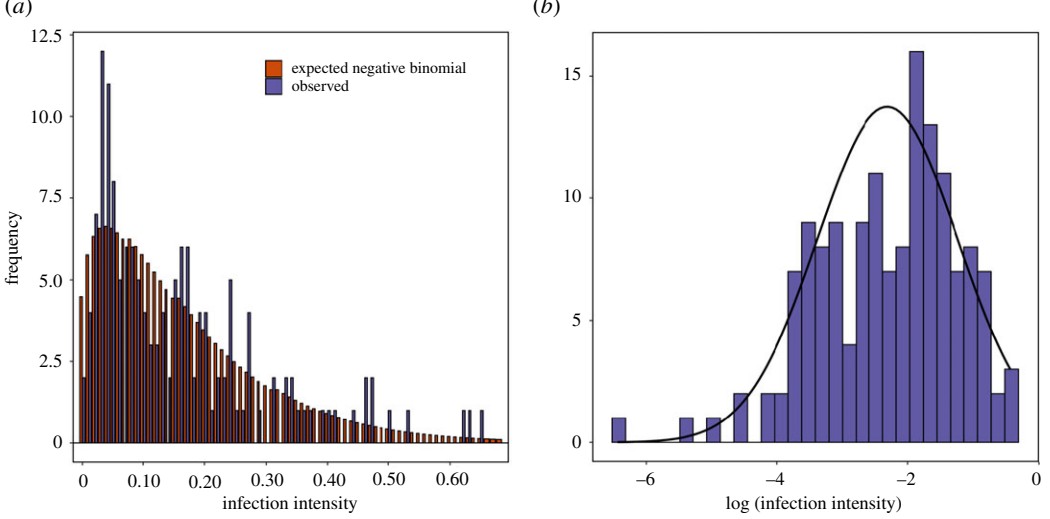

**Figure 1.** (a) Histogram of observed infection intensity (purple bars) for 148 little brown bats exposed to one individual infected with UV fluorescent powder for 24 h. Orange bars represent the expected distribution of infection intensity based on a negative binomial distribution; (b) Histogram of log-transformed infection intensity fit to a normal distribution.

(exploration), $PC1_Y$ (activity) and $PC2_Y$ (sociability) of uninfected hosts, along with $T_{\text{a-dawn}}$ and interactions between sex and all personality traits as predictor variables. We also included trial number for each individual as a random effect to avoid pseudo-replication [49]. To assess variance explained by the fixed and random effects we calculated conditional ($R2_c$) and marginal ($R2_m$) R2 values [50].

Given that the potential amount of UV-dust available to be transmitted within a given trial was limited by the total amount of UV powder on the originally infected bat, the infection intensity of individuals within each trial was non-independent. We, therefore, used permutation tests to generate a series of simulated null models. Specifically, within each iteration of the permutation test, we swapped observed values of infection intensity among individuals in the trial. This procedure, therefore, removed the relationship between animal personality and infection intensity, while also ensuring the total amount of infection was the same within each trial. Code is available at https://github.com/qwebber/uv-powder.

## 3. Results

We retained the first two principal components from both the hole-board PCA and the Y-maze PCA. The first two components from the hole-board trials explained 59.8% of the variance in the data (electronic supplementary material, table S2). The first component from this analysis ($PC1_H$) was associated with activity-based behaviours and the second component ($PC2_H$) with exploration. The first two components from the Y-maze behavioural trials explained 75.7% of the variance in the data (electronic supplementary material, table S3). The first component of this analysis ($PC1_Y$) was also associated with activity-based behaviours, and the second component ($PC2_Y$) was associated with sociability-based behaviours.

All bats became visibly infected with at least some UV powder over the course of their 24 h infection trials (i.e. prevalence of infection = 100%) so we were not able to test for effects of personality on prevalence. However, infection intensity varied widely among individuals ($0.15 \pm 0.14$, range: 0.001–0.65) and between trials (electronic supplementary material, table S1). The observed distribution of infection did not differ from the expected negative binomial distribution ($\chi^2 = 39.7$, d.f. = 68, $p = 0.99$), where most individuals exhibited low intensities of infection and a few individuals highly infected (figure 1).

In partial support of our first hypothesis, more exploratory ($PC2_H$) individuals were more likely to cause higher average infection intensities in the flight tent (table 2; figures 2 and 3). $T_{\text{a-dawn}}$ also influenced transmission with higher infection intensities for uninfected bats on cold nights (table 2 and figure 2), but there was no effect of activity ($PC1_H$ or $PC1_H$), or sociability ($PC2_Y$) on transmission (table 2 and figure 2).

In partial support of our second hypothesis, personality predicted infection acquisition for uninfected individuals and the sex of the uninfected individuals was also important (figure 2). We detected an

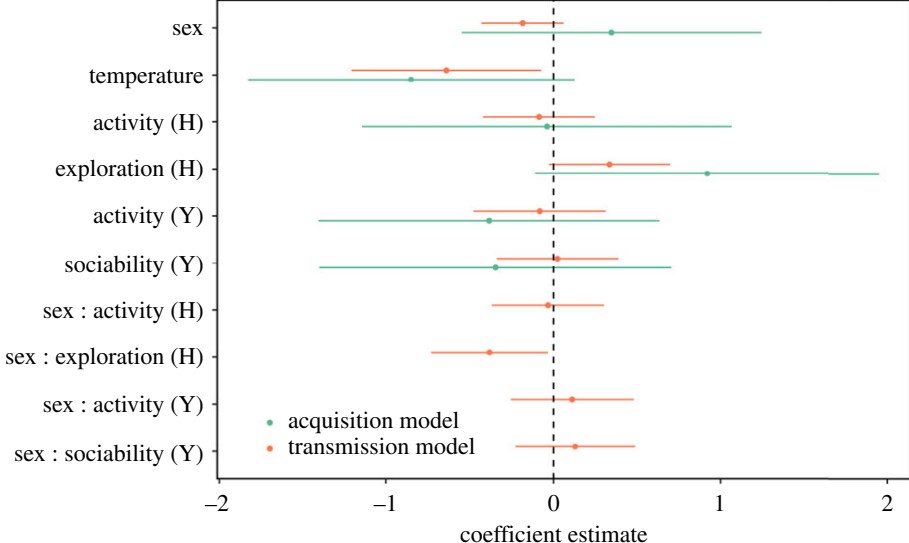

**Figure 2.** Graphical depiction of effect sizes (model coefficients) extracted from the acquisition model (linear model, table 2) and transmission model (linear mixed model, table 3), predicting acquisition and transmission of infection with UV-fluorescent powder for little brown bats.

**Table 2.** Summary of a linear regression model predicting the effect of behaviour of a single infected individual on transmission of a proxy pathogen throughout a group of uninfected individuals (adjusted-$R^2 = 0.75$). The response variable was average acquisition of UV fluorescent powder for uninfected bats from single infected little brown bat over of 10 trials (figure 2 for graphical depiction of effect sizes). Note: reference category for sex was female bats. Bold $p$-values are those whose coefficients do not overlap zero.

|  | coefficient ± s.e. | $t$-value | $p$-value |
| --- | --- | --- | --- |
| intercept | −0.25 ± 0.47 | −0.53 | 0.63 |
| sex | 0.34 ± 0.29 | 1.24 | 0.30 |
| $PC1_H$ (activity) | −0.02 ± 0.21 | −0.11 | 0.92 |
| $PC2_H$ (exploration) | 0.41 ± 0.14 | 2.84 | **0.06** |
| $PC1_Y$ (activity) | −0.12 ± 0.09 | −1.20 | 0.32 |
| $PC2_Y$ (sociability) | −0.11 ± 0.11 | −1.05 | 0.37 |
| $T_{a\text{-dawn}}$ | −0.09 ± 0.03 | −2.76 | **0.07** |

interaction between exploration and sex with more exploratory females, but not males, more likely to acquire higher infection intensities (table 2; figures 2 and 3). Meanwhile, there was no relationship between activity or sociability and infection intensity (table 2 and figure 3). The risk of acquiring infection was greater when $T_{a\text{-dawn}}$ was lower (table 2 and figure 4).

## 4. Discussion

Our results suggest that personality can affect pathogen dynamics in bats but that this relationship is sex-specific. In partial support of our first hypothesis, more exploratory individuals were responsible for higher infection transmission but only to males and not females. This suggests that variation in the explorative tendencies of infected hosts may predict the potential of different individuals to spread pathogens within groups or populations. We also found partial support for our second hypothesis that personality affects an individual's risk of acquiring pathogens, but, again, this effect was sex-specific. More exploratory females were more likely to acquire higher infection intensities from other females, whereas more sociable males were more likely to acquire higher intensities from other males. However, pathogen acquisition by one sex from the other was not affected by personality. In addition to host-specific behaviour, environmental variation also influenced transmission with a strong effect of $T_{a\text{-dawn}}$ on both transmission and acquisition. This could reflect a role for social thermoregulation in

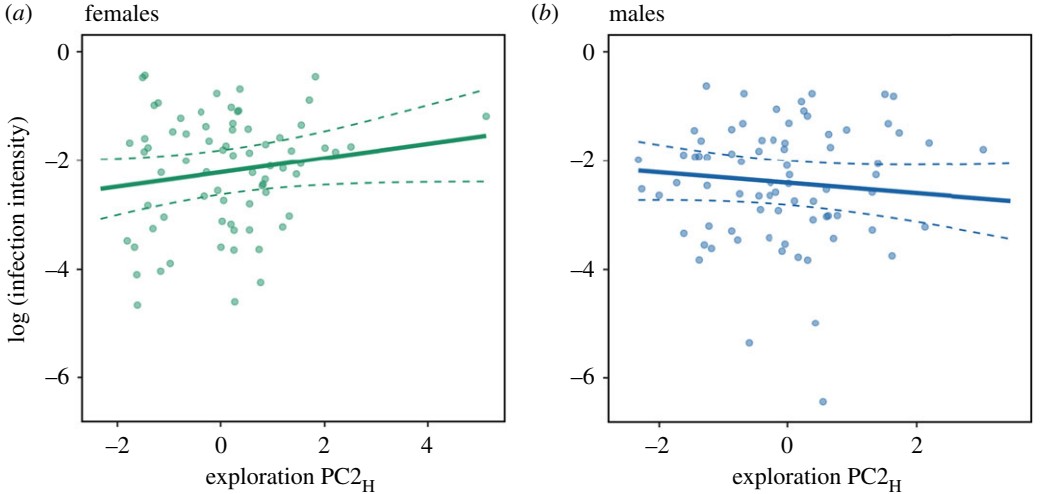

**Figure 3.** Relationship between exploration (PC2$_H$) of the originally infected bat and log-transformed mean infection intensity of originally uninfected little brown bats ($n = 10$ trials) housed in the same flight tent for (*a*) female bats ($n = 76$); and (*b*) male bats ($n = 72$). The interaction between sex and exploration was significant based on bootstrapping the model output (see Material and methods for details) and the trend line was extracted from the mixed model presented in table 3 and dashed lines are 95% confidence intervals around the predicted relationship.

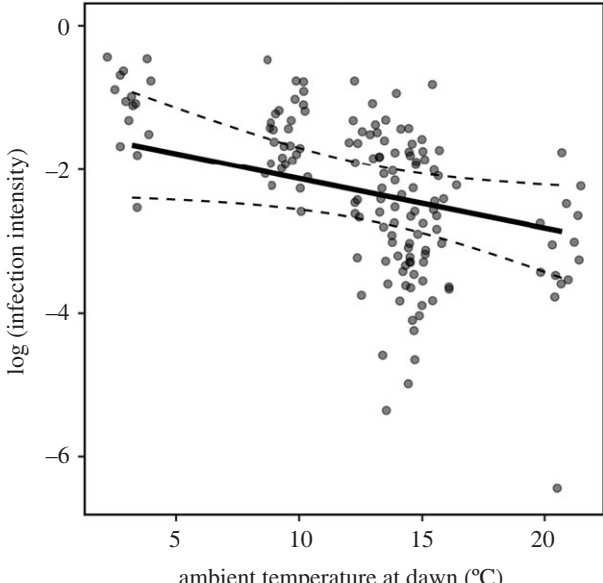

**Figure 4.** Negative relationship between ambient temperature at dawn (°C) and log-transformed infection intensity of UV fluorescent powder for little brown bats ($n = 148$) housed in the experimental flight tent for 24 h. Solid black line was extracted from the global linear mixed model (table 3) and dashed lines are 95% confidence intervals around the predicted relationship. Note: vertical clusters of points are jittered for ease of interpretation and reflect infection data from a single trial.

pathogen transmission for bats and other huddling species. Taken together, results of our experimental manipulation of group composition paint a complex picture highlighting how host sex and behaviour, and environmental variation, can influence pathogen transmission and acquisition.

Behaviour of infected hosts is known to influence how epidemics unfold when infected individuals are introduced to groups of naive but susceptible hosts [6,8]. Quantifying variation in the behavioural composition of groups of bats enabled us to assess the role of host behaviour on transmission in a way which may not be possible under natural conditions. We were not surprised that more exploratory individuals were responsible for higher average infection intensity among uninfected bats. Little brown bats mate promiscuously during autumn swarming with many interactions between males and females [25,30]. More exploratory individuals should, therefore, be responsible for higher

**Table 3.** Summary of the global linear mixed model predicting acquisition of infection with UV-fluorescent powder ($R2_m = 0.13$ and $R2_c = 0.45$). Sex and all personality traits in these models refer to traits of the uninfected individuals exposed to a single infected individual in each trial. Permuted $p$-values represent the probability the observed coefficient differed from a null distribution of coefficients generated based on swapping infection intensities among individuals in the same trial. Note: reference category for sex is females. Bold $p$-values are those whose coefficients do not overlap zero.

| fixed effects | observed coefficient ± s.e. | $t$-value | permuted $p$-value |
|---|---|---|---|
| intercept | 1.30 ± 0.47 | −2.76 | — |
| sex | −0.18 ± 0.13 | −1.46 | 0.10 |
| $PC1_H$ (activity) | −0.02 ± 0.05 | −0.50 | 0.62 |
| $PC2_H$ (exploration) | 0.14 ± 0.07 | 1.82 | **0.03** |
| $PC1_Y$ (activity) | −0.02 ± 0.06 | −0.40 | 0.48 |
| $PC2_Y$ (sociability) | 0.01 ± 0.08 | 0.14 | 0.48 |
| $T_{a\text{-}dawn}$ | −0.07 ± 0.03 | −2.21 | 0.27 |
| sex : $PC1_H$ (activity) | −0.01 ± 0.07 | −0.19 | 0.46 |
| sex : $PC2_H$ (exploration) | −0.24 ± 0.11 | −2.15 | **0.008** |
| sex : $PC1_Y$ (activity) | 0.04 ± 0.07 | 0.61 | 0.32 |
| sex : $PC2_Y$ (sociability) | 0.08 ± 0.11 | 0.72 | 0.35 |
| **random effects** | **variance** | | |
| trial | 0.51 ± 0.72 | | |
| residual | 0.20 ± 0.45 | | |

rates of transmission. In house finches (*Haemorhous mexicanus*), shared space-use at feeder sites influenced the likelihood of *Mycoplasma conjunctivitis* acquisition and transmission, which could expedite an epidemic [51]. For bats, if personality traits are correlated with other behaviours, such as movement or space use [42] and these behaviours influence pathogen dynamics [52] then personality-dependent spatial or movement behaviour could affect pathogen transmission.

Some personality traits also affected pathogen acquisition. More exploratory females acquired higher pathogen intensity, while there was no relationship between exploration and pathogen intensity for males. This could reflect differences in the behaviour and natural history of female versus male bats. Female bats are more gregarious than males, forming often large social groups in maternity colonies each summer. Contact rates in roosts are, therefore, probably higher between females than between males and females. Female bats also appear to be more selective about day roosts than males [53] and the most exploratory females, therefore, may visit, and explore, different roosts more frequently [42]. As a result, exploratory females could be exposed to more conspecifics and more contaminated substrates than less exploratory females resulting in higher infection intensity. That there was no effect of exploration on pathogen intensity for male little brown bats was not necessarily surprising because male bats generally tend to be less social than females in general [29,30]. Our empirical evidence that individual behavioural variation influenced pathogen acquisition supports numerous recent theoretical studies highlighting the importance of behavioural variation in studies of pathogen dynamics [9,54,55].

Our general findings are consistent with theoretical studies [11,15] as well as studies of ectoparasite dynamics for free-ranging bats [16] that suggest more exploratory and sociable individuals should host greater parasite or pathogen loads. An additional implication of our findings relates to the evolution of personality in populations where host survival and fitness is affected by pathogens like *P. destructans*. If more exploratory or sociable individuals disproportionately acquire higher pathogen loads (i.e. a higher dose at the time of initial infection, or repeated doses over time), and potentially face higher mortality as a result, this could alter frequency distributions of personality traits in surviving populations. If personality traits like exploration and sociability are heritable [56] then subsequent generations may be largely composed of individuals with similar traits, and reduced behavioural diversity within populations could place them at risk of stochastic events [57]. Although potential for pathogen-induced directional selection is speculative, patterns of directional selection have been observed as a result of novel predators selecting for prey with specific behavioural traits [58]. For bats, Auteri & Knowles [59] recently reported evidence of selection by WNS on genes thought to influence

vocalizations in bats that have been linked to social behaviour in other species suggesting potential behavioural evolution in response to WNS mortality.

Our results also highlight the importance of environmental variation, and its influence on host behaviour, as a driver of pathogen dynamics. $T_a$ had a strong influence on both pathogen transmission and acquisition, but the direction of the relationship was unexpected. We might have predicted that bats would rely more heavily on torpor during cold nights [60], exhibiting reduced activity and rates of contact with new individuals and, therefore, reduced infection intensities on cold nights. Instead we found that, on colder nights, infection intensities were higher than on warmer nights. Night-roosting social interactions on cold nights could facilitate greater pathogen transmission as bats assess potential roost sites and huddling partners to, presumably, reduce thermoregulatory costs. On colder nights, bats may have been more likely to interact with more individuals and infected substrates, thus increasing their risk of transmission and acquisition [61].

Our findings could be important for understanding pathogen transmission dynamics from both conservation and public health perspectives. Bats in North America currently face an infectious disease of urgent conservation concern, WNS [22]. WNS is caused by the fungal pathogen *P. destructans* which grows in the wing membranes of hibernating bats and causes premature depletion of energy reserves during winter [62,63]. To date, behavioural mechanisms influencing transmission of *P. destructans* remain unknown, although differences in species-specific social behaviour during hibernation appear to play a role [27,64,65]. Our results could have implications for risk of transmission and acquisition of *P. destructans* during autumn swarming and, taken with Auteri & Knowles' [59] evidence of selection on genes potentially associated with sociality, suggest the potential for evolutionary change in behaviour of bats due to WNS. Our results also have implications for understanding the host–pathogen dynamics of potentially zoonotic pathogens occurring in bat populations. We recommend that future studies examine the role of personality and host behaviour in the dynamics of bat viruses like *Myotis lucifugus* coronavirus (Myl-CoV), which occurs naturally in little brown bat populations [28], and for which viral replication is known to increase when hosts are stressed by infection with *P. destructans* [66].

Using captive bats, and a proxy pathogen, allowed us measure personality traits, recapture the same individuals, and assess transmission, in ways that would have been very difficult with free-ranging bats (but see [67]), although our approach does have limitations. Most importantly, the flight tent we used was an artificially small environment where bats were forced into closer contact than they would probably experience in the wild. The day-roosting behaviour of little brown bats during autumn swarming is unknown and, although unlikely for females, it is possible we forced bats to roost together at a time of year when they may have roosted solitarily or in smaller groups. However, the fact that we provided four roosts for only 16 bats per trial, and that bats appeared more likely to roost together on cold nights, suggest they were able to make roosting decisions approximating natural circumstances. Moreover, the negative binomial distribution of infection intensity we observed was almost identical to the pattern observed for natural parasites of free-ranging bats (figure 1, [18]) and pathogens and parasites of animal hosts in general [68]. A critical next step will be to apply our experimental approach to free-ranging bats, possibly by using bio-logging techniques such as PIT tags, radio-telemetry and/or proximity sensors.

In natural host–pathogen systems it may be difficult to assess effects of a given trait on host–pathogen dynamics. For instance, environmental factors, such as $T_a$, can alter virulence or transmission of pathogens [69], while physiological and immunological responses to infection can also influence infection dynamics [70]. Our use of UV powder as a proxy for a pathogen allowed us to isolate the effects of host behaviour from possible effects of $T_a$ on the pathogen or a possible host immune response. Moreover, for bats, UV powder may reliably reflect certain kinds of infections, such as the early stage of invasion of a colony or hibernaculum, by slow-growing *P. destructans*, which could take days or weeks to elicit host behavioural changes. It could also mimic infection with some viruses, to which bats appear to suppress or mitigate immune responses that underlie host behavioural change in other mammals [36,37]. Whether or not UV powder mimics natural infection dynamics, controlling for potential environmental or physiological effects on both the host and pathogen allowed us to examine the direct impact of personality on the trajectory of an epidemic.

# 5. Conclusion

We identified exploration and sociability as personality traits mediating the likelihood of pathogen transmission and acquisition among individual bats. By experimentally manipulating group

composition and quantifying a key environmental variable (i.e. $T_{\text{a-dawn}}$) we were able to identify behavioural and environmental conditions as factors regulating host–pathogen dynamics. Although there will be significant logistical challenges for many bat species, we recommend future experimental work to examine the effects of personality on pathogen transmission for free-ranging bats. Predicting pathogen dynamics is complex, and, while host behaviour may be important, our results also highlight the potential influence of sex-specific differences in behaviour on pathogen dynamics. Future work should integrate the role of animal personality with behavioural measures derived from social network analyses [71] as predictors of pathogen dynamics. As highlighted by Anderson & May [72] in their seminal paper, one of the four principal factors of disease behaviour is: 'the necessity of transmission from one host to the next'. We explored the role of host behaviour as a necessary component of transmission among hosts. While our findings highlight the role of behaviour, future research should focus on the interplay between individual variation in behaviour, physiology, immunology and energetics as necessary components of transmission among hosts.

Ethics. All procedures were approved by the University of Winnipeg Animal Care Committee, conducted in compliance with guidelines of the Canadian Council on Animal Care and approved under Manitoba Conservation Wildlife Scientific Permit number WB16368. Although our study site was negative for *Pseudogymnoascus destructans*, the fungal pathogen that causes white-nose syndrome (WNS), we followed US Fish and Wildlife Service and Canadian Wildlife Health Cooperative (CWHC) guidelines for decontamination by researchers.

Data accessibility. Data and code are archived at: https://zenodo.org/record/3985180#.X1nGnSnPzcs.

Authors' contributions. Q.M.R.W. collected data and performed data analysis, and Q.M.R.W. and C.K.R.W. designed experiments, wrote and edited the paper.

Competing interests. We declare we have no competing interests.

Funding. Funding was provided by a Discovery Grant to C.K.R.W. from the Natural Sciences and Engineering Research Council (NSERC, Canada) and a Manitoba Graduate Scholarship to Q.M.R.W. Q.M.R.W. is currently funded by a Vanier Canada Graduate Scholarship.

Acknowledgements. We are grateful to D. Baloun, E. Low, and H. Mayberry for help with fieldwork. We thank Q. Fletcher, C. Garroway, S. Forbes, and G. Avila-Sakar for outstanding suggestions on earlier versions of this manuscript as well as two anonymous reviewers, one of whom generously provided code to conduct permutation tests. We also thank Manitoba Conservation and P. Ewashko for providing lodging in the field and we respectfully acknowledge that our study took place on and near the traditional territories of Fisher River and Peguis First Nations of Treaty 2.

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
