## [Reviewer comments · Royal Society Open Science]

Review History

RSOS-200770.R0 (Original submission)

Review form: Reviewer 1 (Gerald Carter)

Is the manuscript scientifically sound in its present form?

No

Are the interpretations and conclusions justified by the results?

No

Is the language acceptable?

Yes

Do you have any ethical concerns with this paper?

No

Have you any concerns about statistical analyses in this paper?

Yes

Recommendation?

Major revision is needed (please make suggestions in comments)

Comments to the Author(s)

This study first measured personality (activity, exploration, and gregariousness) of individual bats in 10 captured groups. Then, to simulate a pathogen spreading, the researchers dusted one of the bats with powder and measured how much powder ended up on all the others in the group after 24 h. Their prediction was that more active, exploratory or social bats would transmit and/or acquire more of the simulated pathogen.

Unfortunately, I do have many complaints about this paper, but I think all of them can be addressed. I hope this study is ultimately published and I believe it could be useful.

In my opinion, it has three major flaws, which I think could be fixed.

Problem # 1 Statistical analysis

There are way too many models and inferences (I counted 45 p-values!) for an experiment with 10 trials. First, with only 10 trials, there is going to be very high type 2 error (false negatives) because of low statistical power. So the authors want to do the most powerful test possible, rather than subset their data, and do lots of weak tests with sample sizes of 5 individuals.

Second, when they test so many models, it strongly increases type 1 error (false positives). This is a pretty simple small-scale experiment (only 10 trials), and ideally they should test one clear hypothesis. Instead, they test many combinations of effects: activity measure 1, activity measure 2, exploration 1, exploration 2, in many ways: male to males, males to females, females to females, and so on. They end up with ~40 coefficient estimates and p-values, more p-values than experimental trials! This is a clear example of (unintentional) data dredging. Basically, by testing the hypothesis in too many different ways, the reader is not going to be convinced of any evidence they find. At the very least, they need to control for multiple comparisons using a sequential Bonferroni or something similar.

Another source of type 2 error: they use PCA to get these orthogonal “personality traits” (activity and exploration), but then they add another measure of the same trait (“activity #2”), which should be highly correlated (it even has the same name) and defeats the whole purpose of using PCA to creating independent composite variables. When they add the same variable into a model twice, they make it harder to detect the effect of either one (collinearity). They should create one measure of each personality dimension per bat and re-fit the model.

Also, most of the model assumptions are violated, because they don't have independent observations within trial (for testing effects of personality of infected bats). If bat 1 has more powder that means bat 2 in the same trial has less powder (because powder is limited). I provide a solution to that issue below.

I also looked briefly at the R code and I think the way they derived p-values in the linear mixed models is anti-conservative. I would suggest they use the package `lmerTest` to get good p-values.

Rather than drawing 40 different inferences from many different linear models (each with extreme type 2 error), I think it would be better to select and test one main model that includes all the data. This is shown only in Table S6, but it has problems as outlined above. There is no reason to subset the data by sex unless they detect an interaction effect or have strong reason to think there must be one.

The hypothesis is that the transmission intensity should be affected by both the source bat and the receiver. So one approach would be to construct the model to predict the infection intensity of every receiver bat (but not the source bat).

In this model, the receiver's infection intensity (response) is a function of the receiver's personality, the source bat's personality, the sex composition of actor and receiver (MM, FF, FM), and the temperature. Trial is a random effect.

Personality here could be three independent PCs (like activity, exploration, and sociality) but not correlated measures (like activity 1 and activity 2). I would also present plots of each of these variables as sole predictors of infection intensity. They could also use AIC for model selection. Or perhaps personality could be just one variable that captures activity.

My suggestion (not a requirement just trying to be helpful):

You can use a permutation test to deal with the non-independence in your data. Rather than use degrees of freedom to get your p-value, create a null model to simulate the null hypothesis. For example, in your null model you can swap the infection intensity among the receivers within each trial to remove the relationship between receiver personality and infection density (while keeping the total amount of infection the same within each trial).

Here's some r code as an example. Definitely check this over carefully, because I'm not testing it with any data. It's just to give you an idea of how to do it.

```
# get your data (each row is a unique receiver and source bats are removed)
# y variable is receiver infection intensity
data <- YOUR_DATA

# fit linear mixed model using data
fit <-summary(lmer(YOUR MODEL HERE, data= data))

# get the coefficients you want to test (get p-values for)
obs1 <- fit$coefficients[2,1]
obs2 <- fit$coefficients[3,1]

# now make null model (which makes the null hypothesis true in your data while keeping the
structure realistic)

perms=1000
exp1 <- rep(NA, perms)
exp2 <- rep(NA, perms)

for (i in 1:perms){
  # randomize receiver infection intensities to different receiver bats within each trial
  data$fake.intensity <- as.vector(sapply(by(data$intensity, data$trial, sample),identity))

  # fit model using fake y data
  rfit <-summary(lmer(fake.intensity ~ REST OF YOUR MODEL HERE, data= data))

  # get the coefficient you want to test
  exp1[i] <- rfit$coefficients[2,1]
  exp2[i] <- rfit$coefficients[3,1]
}

# get one-sided p-values by comparing expected and observed

# probability that observed coefficient is greater than expected by chance, assuming null H
mean(exp<=obs)
# probability that observed coefficient is less than expected by chance
mean(exp>=obs)
```

```
# plot permutation test results (red is observed, blue is expected)----
library(ggplot2)
ggplot()+
  geom_histogram(aes(x=exp), color="black",fill="light blue")+
  xlim(min= min(c(exp,obs)), max= max(c(exp,obs)))+
  geom_vline(aes(xintercept=obs), color="red", size=1)+
  xlab("expected values from null model")+
  ggtitle(paste("expected greater than observed: p",
ifelse(mean(exp>=obs)==0,paste("<",1/perms), paste("=",signif(mean(exp>=obs),digits=2))),",",
permutations=",perms, sep=""))

# hope that helps-----
```

Problem #2 The null hypothesis is already way too unlikely

I think this experiment is valuable but not for the reason described in the paper. Let's start with this question: Are more active animals more likely to spread and acquire a disease? Of course. Do we need a demonstration of that fact? Is there any chance that differences in movement rate has no causal influence on social transmission in the real world? I don't think so; the question is only the effect size and if it is large enough to matter given a set of ecological circumstances. The paper presents this as a hypothesis: that IF the personality trait of "activity" is correlated with movement and space use (which it is by definition), and IF these movement affects pathogen transmission (which of course it does), then this means personality COULD impact pathogen transmission...

But logically and mathematically, the effect of variation in movement rates on pathogen transmission cannot be zero. To the extent that personality predicts behavior at all (which is what it means to have personality), there must be an effect of being more active and more social on pathogen transmission (if transmission is by physical contact). I would say that this is an established fact. Every person should still believe this fact to be true regardless of what is reported in this paper.

So this is really an attempt at a demonstration. Fine. But when I see a table of insignificant effects, I just see a failure to detect an effect that we already know exists. The paper says "there was no effect of activity or sociability on infection transmission" but it is only talking about failing to detect the effect in the sample, not the effect in the population.

You might not detect the effect, but it essentially must be there. This is, by the way, how I feel about studies "discovering" personality in species X (or not). With the proper statistical power, one can detect personality in clonal fish raised in identical tanks since birth. I think there is no reason to test the null hypothesis anymore that individuals should be identical in their behavior. Similarly, the idea that host behavior affects pathogen transmission should be the null hypothesis.

The valid question is then not "Does variation in movement \square variation in pathogen transmission?" but rather questions like "How good are the correlations between these measures?" "How well can one predict an individual's propensity to get a disease using standardized test of movement?" Or "If we capture some bats and measure their activity in these 'personality tests', how easy is it to infer their relative impact in a wild contact network?"

In other words, rather than trying to demonstrate, or infer the existence or absence of, these effects which we basically know to be true, it would be better to try to best measure the effect size estimates and their precision. These analyses could be used in meta-analyses and would be helpful for other people trying to measure the link between animal personality measures and outcomes like pathogen transmission using similar kinds of experiments. It could also help people with experimental design. How many bats do I need in my test? How variable is their

behavior likely to be? In my mind, this is what the paper is about (or should be about). It's not about "discovering" that host behavior is repeatable and affects pathogen dynamics. That's a primary assumption of most or all disease ecologists, and I also don't think the reader can infer that at least from the analysis they did.

Problem # 3 Presentation

In my opinion, the scope of this paper and importance of the results is exaggerated way too much. For example, the first sentence reads,

"Given the current global pandemic of coronavirus disease 2019 (COVID-19), it is an understatement to say that infectious diseases of wildlife have become a major concern for human public health"

Is COVID-19 really the relevant place to start? The obvious place to start is with white-nose syndrome, the important disease that is wiping out these bats.

Several points concluded in the discussion cannot be inferred from the results in this paper:

The paper doesn't present a test of an interaction effect of sex (i.e. that personality effect on infection differs by sex). Detecting an effect in only one sex is not the same thing as the slopes being different. The other inferences are not reliable due to problems described above.

There are some confusing arguments made based on switching the meaning of social and sociability. The text needs to use these terms consistently.

It says that the study would be "impossible with free-ranging bats" which is not true. Here's one example: Bakker et al. 2019. Fluorescent biomarkers demonstrate prospects for spreadable vaccines to control disease transmission in wild bats. *Nature Ecology and Evolution*.

There are also some excellent point discussed, such as, that more active/social bats might harbor more parasites, that pathogens can shape personality traits, and that temperature can lead to huddling and change contact networks.

The conclusion statement really oversells this study, and I think the text comes off as grandiose and hyperbolic which really detracts from the whole paper. I would completely delete or rewrite the conclusion. The text says: "Our results are among the first to experimentally manipulate environmental conditions such that the role of host behaviour could be identified as a potential mechanism regulating host-pathogen dynamics." Putting aside the issues with the analysis, why would host behavior not have potential to regulate host-pathogen dynamics? What environmental conditions did they manipulate? It also says: "We also provide a starting point for future work to inform predictions about how host behaviour affects pathogen transmission in natural systems." (??) In my view, the effect of host behavior on pathogen transmission is a large field. For example, look at recent reviews and papers by Matthew Silk, Daniel Streicker, Sonia Altizer, then there's all the works on insects as vectors, etc. That's all before you get to human epidemiology! The classic paper on superspreaders is 15 years old now. The paper says it "clearly highlight the role of demography and environmental conditions" in pathogen dynamics (??). How? The experiment is with 10 captured groups of bats in a tent, where one has UV dust. Having two sexes becomes "demography", having a temperature is the "environment", and the powder is the "pathogen". But there are many studies that study actual demography in natural environments with real pathogens, so it seems strange to bring up these factors here.

When people study social network impacts on disease transmission (like effects of centrality) in a population that is socially structured (rather than spatially structured), it is assumed that these network measures are variable across individuals and repeatable within individuals, so this is the same thing as saying "personality" in different words. That's one of the reasons this idea is not

particularly novel in my opinion; it is already assumed by social network models of disease transmission.

Basically, the tone and scope of the discussion needs to be changed to match the scope of the results. This is not a paper that should change anyone's fundamental understanding of how host behavior might impact pathogen dynamics. It's not an impressive study in disease ecology. But that said, I think it is a neat experiment and could be a useful paper.

Although not a completely novel idea, I so like the idea of discussing behavior and individual variation in behavior as a factor in pathogen transmission, and I think there is much good work to do on that topic (which is one of the reasons this study provides useful data).

Finally, it would be better to present all the relevant scatterplot correlations (in the supplement), not only the ones that were significant.

Summary

I'm sorry to be so negative. If my tone seemed disrespectful, rude or unkind, please know that was not my intention. Despite the problems as written now, this paper has useful information in it that I might even use in my own research. For example, the data here would help me do a power analysis when designing similar experiments on different questions. I hope my extensive criticism does not discourage the authors and that my suggested revisions improve the paper.

Other minor line by line comments

Regarding the title, I would discourage the use of the phrase "proxy pathogen". To me that sounds like a tractable or safer pathogen used to study another less tractable pathogen. Also the use of transmission and acquisition in the title is redundant because you cannot increase just one of them without increasing the other.

I would just say "Evidence that personality affects pathogen transmission in little brown bats" or something similar (if they indeed can find that).

Abstract (and also maybe the methods) needs to explain experimental design better. Range of bats in each group? How many groups? How many trials? Are any bats used in multiple groups?

Line 61. Is personality a behavioral "mechanism"? Not in my mind. I would call personality a behavioral phenotype or a factor that affects transmission. It describes proclivity towards behaviors that determine how often or how quickly a pathogen could spread, but it does not explain how transmission works. For example, compassion is not a "mechanism" of violence, but it is a factor that might influence whether violence occurs or not.

Line 68. *Many* bats are highly gregarious (not all)

Rephrase for clarity: "social structure appears ephemeral"

Line 93. You should say here that you captured and tested bats that were swarming to link to the above text.

Line 97. What does "naïve" mean here? If it means "uninfected", then maybe just say that.

Be clear that your experiment is not measuring who gets what from whom. A bat could acquire the pathogen directly from the experimentally infected bat or indirectly from another bat or the enclosure wall, right? So I would say that the second hypothesis is that the more active bats are more likely to acquire the pathogen (but not specify how or from whom).

Line 149. Why unique? There are many studies on how variation among individuals affects infection dynamics.

Line 182. An important aspect of measuring personality is putting animals in a standard environment. In one trial, the stimulus bat might call with echolocation or even social calls and in another trial it might be silent. The behavior of the stimulus bat might matter more than the personality of the focal bat, so this is likely to be a noisy measure of sociality (and might be worth mentioning this as a caveat).

I think Table S2 should be in the main text (if the journal allows this amount of space).

Line 240 Code is available...

Line 293 typo : 'that'

369 "The negative binomial distribution of infection intensity we observed was almost identical to the pattern observed for natural parasites of free-ranging bats" If this is true (almost identical), then this is maybe the most interesting part of the paper in my opinion. Please present the model fit and an analysis!

Review form: Reviewer 2

Is the manuscript scientifically sound in its present form?

Yes

Are the interpretations and conclusions justified by the results?

Yes

Is the language acceptable?

Yes

Do you have any ethical concerns with this paper?

No

Have you any concerns about statistical analyses in this paper?

No

Recommendation?

Accept with minor revision (please list in comments)

Comments to the Author(s)

This is a very nicely designed and carried out study, able to explore the causality of personality in explaining variation in pathogen spread. The paper is clearly written, analyses sound and results soundly discussed. I only have 2 v minor comments:

L66: I suggest that this reference is of relevance here:

Petkova et al (2018). Parasite infection and host personality: Glugea infected three-spined sticklebacks are more social. *BEHAVIORAL ECOLOGY AND SOCIOBIOLOGY* 72:173.
doi:10.1007/s00265-018-2586-3.

L151-: it is not clear that only 1 individual per group was infected by the text here, neither how this was chosen (in terms of personality). Also, was personality assessed prior or after observations? Jus double check that these aspects are clear.

Decision letter (RSOS-200770.R0)

Dear Mr Webber,

The editors assigned to your paper ("Personality affects transmission and acquisition of a proxy pathogen in little brown bats") have now received comments from reviewers. We would like you to revise your paper in accordance with the referee and Associate Editor suggestions which can be found below (not including confidential reports to the Editor). Please note this decision does not guarantee eventual acceptance.

Please submit a copy of your revised paper before 24-Jun-2020. Please note that the revision deadline will expire at 00.00am on this date. If we do not hear from you within this time then it will be assumed that the paper has been withdrawn. In exceptional circumstances, extensions may be possible if agreed with the Editorial Office in advance. We do not allow multiple rounds of revision so we urge you to make every effort to fully address all of the comments at this stage. If deemed necessary by the Editors, your manuscript will be sent back to one or more of the original reviewers for assessment. If the original reviewers are not available, we may invite new reviewers.

- Data accessibility

If you wish to submit your supporting data or code to Dryad (<http://datadryad.org/>), or modify your current submission to dryad, please use the following link:
<http://datadryad.org/submit?journalID=RSOS&manu=RSOS-200770>

- Competing interests

- Authors' contributions

- Acknowledgements

- Funding statement

Kind regards,

Andrew Dunn

on behalf of Dr Kimberley Mathot (Associate Editor) and Pete Smith (Subject Editor)

Associate Editor's comments (Dr Kimberley Mathot):

Two reviewers and I have examined the manuscript. We all feel that the manuscript presents potentially interesting results, that with appropriate revisions, may be suitable for publication. Reviewer #1 offers a number of suggestions to improve the manuscript, and reviewer #2 identifies some missing literature and areas for clarification in the text.

I share reviewer #1's concern that the large number of statistical tests mean that there is substantial risk of Type I error, while on the other hand, the small sample sizes mean there is risk of Type II error. Rather than now post-hoc choosing a subset of analyses to present, as referee #1 suggests, I would instead ask that the authors focus on effect sizes and uncertainty when

interpreting their findings (both non-significant and significant). This will address both types of error, without running the risk that knowing the results of the multiple analyses influences the choice of a subset of models going forward. This would also move towards addressing reviewer #1's suggestion that whether or not there is an effect of host behaviour on transmission is a less interesting question than the magnitude of the effect.

Referee #1 also raises some concern about the use of two separate synthetic variables to capture the latent variable "activity". I don't share this concern. The authors explicitly state that they tested for collinearity between variables, which I interpret to mean that there was no collinearity between activity 1 and activity 2. This type of pattern is commonly found in animal personality work (i.e., two separate assays appear to measure the same trait, but are themselves uncorrelated). However, as you do not directly quantify repeatability in the current study, please provide the repeatability ranges you refer to in lines 171-173 based on previously published studies.

I also agree with referee #1 that there are areas of the text that should be revised. Specifically, that framing this paper in terms of white-nose syndrome would be more appropriate than the current COVID 19 framing, and also several areas in the conclusions where the significance of the study is overstated.

Based on my own reading, and the reports from the two referees, I would recommend major revisions.

Comments to Author:

Reviewers' Comments to Author:

Reviewer: 1

Comments to the Author(s)

This study first measured personality (activity, exploration, and gregariousness) of individual bats in 10 captured groups. Then, to simulate a pathogen spreading, the researchers dusted one of the bats with powder and measured how much powder ended up on all the others in the group after 24 h. Their prediction was that more active, exploratory or social bats would transmit and/or acquire more of the simulated pathogen.

Unfortunately, I do have many complaints about this paper, but I think all of them can be addressed. I hope this study is ultimately published and I believe it could be useful.

In my opinion, it has three major flaws, which I think could be fixed.

Problem # 1 Statistical analysis

There are way too many models and inferences (I counted 45 p-values!) for an experiment with 10 trials. First, with only 10 trials, there is going to be very high type 2 error (false negatives) because of low statistical power. So the authors want to do the most powerful test possible, rather than subset their data, and do lots of weak tests with sample sizes of 5 individuals.

Second, when they test so many models, it strongly increases type 1 error (false positives). This is a pretty simple small-scale experiment (only 10 trials), and ideally they should test one clear hypothesis. Instead, they test many combinations of effects: activity measure 1, activity measure 2, exploration 1, exploration 2, in many ways: male to males, males to females, females to females, and so on. They end up with ~40 coefficient estimates and p-values, more p-values than experimental trials! This is a clear example of (unintentional) data dredging. Basically, by testing the hypothesis in too many different ways, the reader is not going to be convinced of any evidence they find. At the very least, they need to control for multiple comparisons using a sequential Bonferroni or something similar.

Another source of type 2 error: they use PCA to get these orthogonal "personality traits" (activity and exploration), but then they add another measure of the same trait ("activity #2"), which

should be highly correlated (it even has the same name) and defeats the whole purpose of using PCA to creating independent composite variables. When they add the same variable into a model twice, they make it harder to detect the effect of either one (collinearity). They should create one measure of each personality dimension per bat and re-fit the model.

Also, most of the model assumptions are violated, because they don't have independent observations within trial (for testing effects of personality of infected bats). If bat 1 has more powder that means bat 2 in the same trial has less powder (because powder is limited). I provide a solution to that issue below.

I also looked briefly at the R code and I think the way they derived p-values in the linear mixed models is anti-conservative. I would suggest they use the package `lmerTest` to get good p-values.

Rather than drawing 40 different inferences from many different linear models (each with extreme type 2 error), I think it would be better to select and test one main model that includes all the data. This is shown only in Table S6, but it has problems as outlined above. There is no reason to subset the data by sex unless they detect an interaction effect or have strong reason to think there must be one.

The hypothesis is that the transmission intensity should be affected by both the source bat and the receiver. So one approach would be to construct the model to predict the infection intensity of every receiver bat (but not the source bat).

In this model, the receiver's infection intensity (response) is a function of the receiver's personality, the source bat's personality, the sex composition of actor and receiver (MM, FF, FM), and the temperature. Trial is a random effect.

Personality here could be three independent PCs (like activity, exploration, and sociality) but not correlated measures (like activity 1 and activity 2). I would also present plots of each of these variables as sole predictors of infection intensity. They could also use AIC for model selection. Or perhaps personality could be just one variable that captures activity.

My suggestion (not a requirement just trying to be helpful):

You can use a permutation test to deal with the non-independence in your data. Rather than use degrees of freedom to get your p-value, create a null model to simulate the null hypothesis. For example, in your null model you can swap the infection intensity among the receivers within each trial to remove the relationship between receiver personality and infection density (while keeping the total amount of infection the same within each trial).

Here's some R code as an example. Definitely check this over carefully, because I'm not testing it with any data. It's just to give you an idea of how to do it.

```
# get your data (each row is a unique receiver and source bats are removed)
# y variable is receiver infection intensity
data <- YOUR_DATA
```

```
# fit linear mixed model using data
fit <- summary(lmer(YOUR MODEL HERE, data= data))
```

```
# get the coefficients you want to test (get p-values for)
obs1 <- fit$coefficients[2,1]
obs2 <- fit$coefficients[3,1]
```

```
# now make null model (which makes the null hypothesis true in your data while keeping the structure realistic)
```

```

perms=1000
exp1 <- rep(NA, perms)
exp2 <- rep(NA, perms)

for (i in 1:perms){
  # randomize receiver infection intensities to different receiver bats within each trial
  data$fake.intensity <- as.vector(sapply(by(data$intensity, data$trial, sample),identity))

  # fit model using fake y data
  rfit <-summary(lmer(fake.intensity ~ REST OF YOUR MODEL HERE, data= data))

  # get the coefficient you want to test
  exp1[i] <- rfit$coefficients[2,1]
  exp2[i] <- rfit$coefficients[3,1]
}

# get one-sided p-values by comparing expected and observed

# probability that observed coefficient is greater than expected by chance, assuming null H
mean(exp<=obs)
# probability that observed coefficient is less than expected by chance
mean(exp>=obs)

# plot permutation test results (red is observed, blue is expected)----
library(ggplot2)
ggplot()+
  geom_histogram(aes(x=exp), color="black",fill="light blue")+
  xlim(min= min(c(exp,obs)), max= max(c(exp,obs)))+
  geom_vline(aes(xintercept=obs), color="red", size=1)+
  xlab("expected values from null model")+
  ggtitle(paste("expected greater than observed: p",
ifelse(mean(exp>=obs)==0,paste("<",1/perms), paste("=",signif(mean(exp>=obs),digits=2))),",",
permutations="perms, sep=""))

# hope that helps-----

```

Problem #2 The null hypothesis is already way too unlikely

I think this experiment is valuable but not for the reason described in the paper. Let's start with this question: Are more active animals more likely to spread and acquire a disease? Of course. Do we need a demonstration of that fact? Is there any chance that differences in movement rate has no causal influence on social transmission in the real world? I don't think so; the question is only the effect size and if it is large enough to matter given a set of ecological circumstances. The paper presents this as a hypothesis: that IF the personality trait of "activity" is correlated with movement and space use (which it is by definition), and IF these movement affects pathogen transmission (which of course it does), then this means personality COULD impact pathogen transmission...

But logically and mathematically, the effect of variation in movement rates on pathogen transmission cannot be zero. To the extent that personality predicts behavior at all (which is what it means to have personality), there must be an effect of being more active and more social on pathogen transmission (if transmission is by physical contact). I would say that this is an established fact. Every person should still believe this fact to be true regardless of what is reported in this paper.

So this is really an attempt at a demonstration. Fine. But when I see a table of insignificant effects, I just see a failure to detect an effect that we already know exists. The paper says “there was no effect of activity or sociability on infection transmission” but it is only talking about failing to detect the effect in the sample, not the effect in the population.

You might not detect the effect, but it essentially must be there. This is, by the way, how I feel about studies “discovering” personality in species X (or not). With the proper statistical power, one can detect personality in clonal fish raised in identical tanks since birth. I think there is no reason to test the null hypothesis anymore that individuals should be identical in their behavior. Similarly, the idea that host behavior affects pathogen transmission should be the null hypothesis.

The valid question is then not “Does variation in movement \square variation in pathogen transmission?” but rather questions like “How good are the correlations between these measures?” “How well can one predict an individual’s propensity to get a disease using standardized test of movement?” Or “If we capture some bats and measure their activity in these ‘personality tests’, how easy is it to infer their relative impact in a wild contact network?”

In other words, rather than trying to demonstrate, or infer the existence or absence of, these effects which we basically know to be true, it would be better to try to best measure the effect size estimates and their precision. These analyses could be used in meta-analyses and would be helpful for other people trying to measure the link between animal personality measures and outcomes like pathogen transmission using similar kinds of experiments. It could also help people with experimental design. How many bats do I need in my test? How variable is their behavior likely to be? In my mind, this is what the paper is about (or should be about). It’s not about “discovering” that host behavior is repeatable and affects pathogen dynamics. That’s a primary assumption of most or all disease ecologists, and I also don’t think the reader can infer that at least from the analysis they did.

Problem # 3 Presentation

In my opinion, the scope of this paper and importance of the results is exaggerated way too much. For example, the first sentence reads,

“Given the current global pandemic of coronavirus disease 2019 (COVID-19), it is an understatement to say that infectious diseases of wildlife have become a major concern for human public health”

Is COVID-19 really the relevant place to start? The obvious place to start is with white-nose syndrome, the important disease that is wiping out these bats.

Several points concluded in the discussion cannot be inferred from the results in this paper:

The paper doesn’t present a test of an interaction effect of sex (i.e. that personality effect on infection differs by sex). Detecting an effect in only one sex is not the same thing as the slopes being different. The other inferences are not reliable due to problems described above.

There are some confusing arguments made based on switching the meaning of social and sociability. The text needs to use these terms consistently.

It says that the study would be “impossible with free-ranging bats” which is not true. Here’s one example: Bakker et al. 2019. Fluorescent biomarkers demonstrate prospects for spreadable vaccines to control disease transmission in wild bats. *Nature Ecology and Evolution*.

There are also some excellent points discussed, such as, that more active/social bats might harbor more parasites, that pathogens can shape personality traits, and that temperature can lead to huddling and change contact networks.

The conclusion statement really oversells this study, and I think the text comes off as grandiose and hyperbolic which really detracts from the whole paper. I would completely delete or rewrite the conclusion. The text says: "Our results are among the first to experimentally manipulate environmental conditions such that the role of host behaviour could be identified as a potential mechanism regulating host-pathogen dynamics." Putting aside the issues with the analysis, why would host behavior not have potential to regulate host-pathogen dynamics? What environmental conditions did they manipulate? It also says: "We also provide a starting point for future work to inform predictions about how host behaviour affects pathogen transmission in natural systems." (??) In my view, the effect of host behavior on pathogen transmission is a large field. For example, look at recent reviews and papers by Matthew Silk, Daniel Streicker, Sonia Altizer, then there's all the works on insects as vectors, etc. That's all before you get to human epidemiology! The classic paper on superspreaders is 15 years old now. The paper says it "clearly highlight the role of demography and environmental conditions" in pathogen dynamics (??). How? The experiment is with 10 captured groups of bats in a tent, where one has UV dust. Having two sexes becomes "demography", having a temperature is the "environment", and the powder is the "pathogen". But there are many studies that study actual demography in natural environments with real pathogens, so it seems strange to bring up these factors here.

When people study social network impacts on disease transmission (like effects of centrality) in a population that is socially structured (rather than spatially structured), it is assumed that these network measures are variable across individuals and repeatable within individuals, so this is the same thing as saying "personality" in different words. That's one of the reasons this idea is not particularly novel in my opinion; it is already assumed by social network models of disease transmission.

Basically, the tone and scope of the discussion needs to be changed to match the scope of the results. This is not a paper that should change anyone's fundamental understanding of how host behavior might impact pathogen dynamics. It's not an impressive study in disease ecology. But that said, I think it is a neat experiment and could be a useful paper.

Although not a completely novel idea, I so like the idea of discussing behavior and individual variation in behavior as a factor in pathogen transmission, and I think there is much good work to do on that topic (which is one of the reasons this study provides useful data).

Finally, it would be better to present all the relevant scatterplot correlations (in the supplement), not only the ones that were significant.

Summary

I'm sorry to be so negative. If my tone seemed disrespectful, rude or unkind, please know that was not my intention. Despite the problems as written now, this paper has useful information in it that I might even use in my own research. For example, the data here would help me do a power analysis when designing similar experiments on different questions. I hope my extensive criticism does not discourage the authors and that my suggested revisions improve the paper.

Other minor line by line comments

Regarding the title, I would discourage the use of the phrase "proxy pathogen". To me that sounds like a tractable or safer pathogen used to study another less tractable pathogen. Also the use of transmission and acquisition in the title is redundant because you cannot increase just one of them without increasing the other.

I would just say “Evidence that personality affects pathogen transmission in little brown bats” or something similar (if they indeed can find that).

Abstract (and also maybe the methods) needs to explain experimental design better. Range of bats in each group? How many groups? How many trials? Are any bats used in multiple groups?

Line 61. Is personality a behavioral “mechanism”? Not in my mind. I would call personality a behavioral phenotype or a factor that affects transmission. It describes proclivity towards behaviors that determine how often or how quickly a pathogen could spread, but it does not explain how transmission works. For example, compassion is not a “mechanism” of violence, but it is a factor that might influence whether violence occurs or not.

Line 68. *Many* bats are highly gregarious (not all)

Rephrase for clarity: “social structure appears ephemeral”

Line 93. You should say here that you captured and tested bats that were swarming to link to the above text.

Line 97. What does “naïve” mean here? If it means “uninfected”, then maybe just say that.

Be clear that your experiment is not measuring who gets what from whom. A bat could acquire the pathogen directly from the experimentally infected bat or indirectly from another bat or the enclosure wall, right? So I would say that the second hypothesis is that the more active bats are more likely to acquire the pathogen (but not specify how or from whom).

Line 149. Why unique? There are many studies on how variation among individuals affects infection dynamics.

Line 182. An important aspect of measuring personality is putting animals in a standard environment. In one trial, the stimulus bat might call with echolocation or even social calls and in another trial it might be silent. The behavior of the stimulus bat might matter more than the personality of the focal bat, so this is likely to be a noisy measure of sociality (and might be worth mentioning this as a caveat).

I think Table S2 should be in the main text (if the journal allows this amount of space).

Line 240 Code is available...

Line 293 typo : ‘that’

369 “The negative binomial distribution of infection intensity we observed was almost identical to the pattern observed for natural parasites of free-ranging bats” If this is true (almost identical), then this is maybe the most interesting part of the paper in my opinion. Please present the model fit and an analysis!

Reviewer: 2

Comments to the Author(s)

This is a very nicely designed and carried out study, able to explore the causality of personality in explaining variation in pathogen spread. The paper is clearly written, analyses sound and results soundly discussed. I only have 2 v minor comments:

L66: I suggest that this reference is of relevance here:

Petkova et al (2018). Parasite infection and host personality: Glugea infected three-spined sticklebacks are more social. BEHAVIORAL ECOLOGY AND SOCIOBIOLOGY 72:173. doi:10.1007/s00265-018-2586-3.

L151-: it is not clear that only 1 individual per group was infected by the text here, neither how this was chosen (in terms of personality). Also, was personality assessed prior or after observations? Jus double check that these aspects are clear.

Author's Response to Decision Letter for (RSOS-200770.R0)

See Appendix A.

Decision letter (RSOS-200770.R1)

Dear Mr Webber

On behalf of the Editors, we are pleased to inform you that your Manuscript RSOS-200770.R1 "Personality affects dynamics of an experimental pathogen in little brown bats" has been accepted for publication in Royal Society Open Science subject to minor revision in accordance with the referees' reports. Please find the referees' comments along with any feedback from the Editors below my signature.

Please submit your revised manuscript and required files (see below) no later than 7 days from today's (ie 17-Aug-2020) date. Note: the ScholarOne system will 'lock' if submission of the revision is attempted 7 or more days after the deadline. If you do not think you will be able to meet this deadline please contact the editorial office immediately.

on behalf of Dr Kimberley Mathot (Associate Editor) and Pete Smith (Subject Editor)
 openscience@royalsociety.org

Associate Editor Comments to Author (Dr Kimberley Mathot):

Comments to the Author:

Thank you for submitting the revised version of your manuscript. I'm satisfied that you have addressed the major comments/concerns raised in the initial review, but have a number of very small edits I would like you to make. Several of these relate to your use of the term "personality" to refer to individuals. Personality is a population level characteristic (a measure of the proportion of phenotypic variance in a population that exists at the among-individual level) - not an individual attribute. As such, statements like "the personality of the infected individual" are technically incorrect.

Minor edits

Line 18: please rephrase to "We quantified activity and sociability in little brown bats (*Myotis lucifugus*) and then experimentally...."

Lines 20-21: please rephrase to : "1) more sociable and more exploratory individuals..."

Line 22: please rephrase to : "...more sociable and exploratory...."

Lines 29-31: please rephrase to "As predicted, the exploratory behaviour of the experimentally infected individual was positively correlated with infection intensity in their group-mates, while more"

Lines 56: please rephrase to "Consistent among-individual...."

Line 89: replace "within" with "during".

Line 96: replace "personality" with "behavioural"

Line 97: replace "personality" with "behavioural type"

Line 101: replace "personality" with "behavioural type"

Line 173: rephrase to "....., all personality traits measured here have previously been shown to be weakly to highly repeatable..."

Line 227: replace "personality traits" with "behavioural traits"

Line 292: replace "personality composition" with "behavioural composition" and replace "host personality" with "host behaviour"

Line 348-349: rephrase to "Our finding could be important for understanding pathogen transmission dynamics from both conservation and public health perspectives".

Line 394-395: rephrase to "..... we were able to identify behavioural and environmental..."

Line 398: rephrase to "host behaviour"

Line 636: rephrase to "the effect of behavioural of a single..."

It's of course a personal choice, but I would also recommend acknowledging the referee that signed their review and provided you with sample code.

===PREPARING YOUR MANUSCRIPT===

- one version identifying all the changes that have been made (for instance, in coloured highlight, in bold text, or tracked changes);
- a 'clean' version of the new manuscript that incorporates the changes made, but does not highlight them. This version will be used for typesetting.

===PREPARING YOUR REVISION IN SCHOLARONE===

- Ensure that your data access statement meets the requirements at <https://royalsociety.org/journals/authors/author-guidelines/#data>. You should ensure that you cite the dataset in your reference list. If you have deposited data etc in the Dryad repository, please only include the 'For publication' link at this stage. You should remove the 'For review' link.
- If you are requesting an article processing charge waiver, you must select the relevant waiver option (if requesting a discretionary waiver, the form should have been uploaded at Step 3 'File upload' above).
- If you have uploaded ESM files, please ensure you follow the guidance at <https://royalsociety.org/journals/authors/author-guidelines/#supplementary-material> to include a suitable title and informative caption. An example of appropriate titling and captioning may be found at https://figshare.com/articles/Table_S2_from_Is_there_a_trade-off_between_peak_performance_and_performance_breadth_across_temperatures_for_aerobic_scope_in_teleost_fishes_/3843624.

Author's Response to Decision Letter for (RSOS-200770.R1)

See Appendix B.

Decision letter (RSOS-200770.R2)

Dear Mr Webber,

It is a pleasure to accept your manuscript entitled "Personality affects dynamics of an experimental pathogen in little brown bats" in its current form for publication in Royal Society Open Science.

Due to rapid publication and an extremely tight schedule, if comments are not received, your paper may experience a delay in publication. Royal Society Open Science operates under a continuous publication model. Your article will be published straight into the next open issue and this will be the final version of the paper. As such, it can be cited immediately by other researchers. As the issue version of your paper will be the only version to be published I would

advise you to check your proofs thoroughly as changes cannot be made once the paper is published.

on behalf of Dr Kimberley Mathot (Associate Editor) and Pete Smith (Subject Editor)
openscience@royalsociety.org

Appendix A

Associate Editor's comments (Dr Kimberley Mathot):

Two reviewers and I have examined the manuscript. We all feel that the manuscript presents potentially interesting results, that with appropriate revisions, may be suitable for publication. Reviewer #1 offers a number of suggestions to improve the manuscript, and reviewer #2 identifies some missing literature and areas for clarification in the text.

I share reviewer #1's concern that the large number of statistical tests mean that there is substantial risk of Type I error, while on the other hand, the small sample sizes mean there is risk of Type II error. Rather than now post-hoc choosing a subset of analyses to present, as referee #1 suggests, I would instead ask that the authors focus on effect sizes and uncertainty when interpreting their findings (both non-significant and significant). This will address both types of error, without running the risk that knowing the results of the multiple analyses influences the choice of a subset of models going forward. This would also move towards addressing reviewer #1's suggestion that whether or not there is an effect of host behaviour on transmission is a less interesting question than the magnitude of the effect.

Response: we now run a single "pathogen transmission" model (Table 2) and a single "pathogen acquisition" model (see Table 3). We chose not to incorporate all data into a single model as suggested by reviewer 1 partially because that would require including all 4 personality traits for the both receiver and transmitter bats as well as sex and temperature which is clearly too many variables for our sample size. However, instead, we followed the direction of reviewer 1 and generated p-values based on a permutation of mixed models (for the acquisition model). We also now present a figure showing effect sizes/model coefficients for each variable (Figure 3).

Referee #1 also raises some concern about the use of two separate synthetic variables to capture the latent variable "activity". I don't share this concern. The authors explicitly state that they tested for collinearity between variables, which I interpret to mean that there was no collinearity between activity 1 and activity 2. This type of pattern is commonly found in animal personality work (i.e., two separate assays appear to measure the same trait, but are themselves uncorrelated). However, as you do not directly quantify repeatability in the current study, please provide the repeatability ranges you refer to in lines 171-173 based on previously published studies.

Response: we now refer provide the repeatability range from our previous study (see lines 172-175). We also further clarify our test for collinearity and that all variance inflation factors were <2.2 at lines 222-223.

I also agree with referee #1 that there are areas of the text that should be revised. Specifically, that framing this paper in terms of white-nose syndrome would be more appropriate than the current COVID 19 framing, and also several areas in the conclusions where the significance of the study is overstated.

Response: We have restructured the beginning of the paper (see response to Reviewer 1 below) but we respectfully disagree that WNS is the only context for the study. We think it remains important to highlight that many (in fact most) human diseases are zoonotic and that it is

Commented [W1]: Do any of the conclusions change? It would be nice to be able to say that none of the conclusions changed (or they changed very little) and if they did which ones did and by how much.

important to understand pathogen transmission dynamics for host taxa that are known to be important sources of zoonotic pathogens (lines 64-66). We have, however, revised the conclusion to avoid overstating the significance of the results.

Based on my own reading, and the reports from the two referees, I would recommend major revisions.

Response: thank you and the reviewers for very helpful comments on our manuscript. Please see below for detailed responses to all comments.

Comments to Author:

Reviewers' Comments to Author:

Reviewer: 1

Comments to the Author(s)

This study first measured personality (activity, exploration, and gregariousness) of individual bats in 10 captured groups. Then, to simulate a pathogen spreading, the researchers dusted one of the bats with powder and measured how much powder ended up on all the others in the group after 24 h. Their prediction was that more active, exploratory or social bats would transmit and/or acquire more of the simulated pathogen.

Unfortunately, I do have many complaints about this paper, but I think all of them can be addressed. I hope this study is ultimately published and I believe it could be useful.

Response: thank you for your comments on our manuscript. We have addressed them all in detail below.

In my opinion, it has three major flaws, which I think could be fixed.

Problem # 1 Statistical analysis

There are way too many models and inferences (I counted 45 p-values!) for an experiment with 10 trials. First, with only 10 trials, there is going to be very high type 2 error (false negatives) because of low statistical power. So the authors want to do the most powerful test possible, rather than subset their data, and do lots of weak tests with sample sizes of 5 individuals.

Second, when they test so many models, it strongly increases type 1 error (false positives). This is a pretty simple small-scale experiment (only 10 trials), and ideally they should test one clear hypothesis. Instead, they test many combinations of effects: activity measure 1, activity measure 2, exploration 1, exploration 2, in many ways: male to males, males to females, females to females, and so on. They end up with ~40 coefficient estimates and p-values, more p-values than experimental trials! This is a clear example of (unintentional) data dredging. Basically, by testing the hypothesis in too many different ways, the reader is not going to be convinced of any evidence they find. At the very least, they need to control for multiple comparisons using a

sequential Bonferroni or something similar.

Response: we have restructured our modelling framework. We now run a single model and use the permutation approach proposed below. Importantly, the conclusions have not changed. Exploration remains a significant predictor of infection, as it was in the previous analysis.

Another source of type 2 error: they use PCA to get these orthogonal “personality traits” (activity and exploration), but then they add another measure of the same trait (“activity #2”), which should be highly correlated (it even has the same name) and defeats the whole purpose of using PCA to creating independent composite variables. When they add the same variable into a model twice, they make it harder to detect the effect of either one (collinearity). They should create one measure of each personality dimension per bat and re-fit the model.

Response: as noted by the associate editor, it is common practice in studies of animal personality to estimate behaviours in multiple independent contexts and use these variables in the same model. In this case, the two contexts were hole-board and Y-maze tests which were conducted independent from one another. Furthermore, the use of PCA reduces the behavioural variables measured in the tests into uncorrelated components (i.e. PC1 and PC2 from the hole-board PCA are not correlated). As noted by the associate editor, we also tested for collinearity between variables and found none. Therefore, we still use of these variables in our models.

Also, most of the model assumptions are violated, because they don’t have independent observations within trial (for testing effects of personality of infected bats). If bat 1 has more powder that means bat 2 in the same trial has less powder (because powder is limited). I provide a solution to that issue below.

Response: see below for response to this comment and lines 240-247 of the manuscript.

I also looked briefly at the R code and I think the way they derived p-values in the linear mixed models is anti-conservative. I would suggest they use the package lmerTest to get good p-values.

Response: as noted, we now use the permutation technique to derive p-values.

Rather than drawing 40 different inferences from many different linear models (each with extreme type 2 error), I think it would be better to select and test one main model that includes all the data. This is shown only in Table S6, but it has problems as outlined above. There is no reason to subset the data by sex unless they detect an interaction effect or have strong reason to think there must be one.

Response: as suggested, we now run two separate models for the transmission and acquisition data with all variables and interactions and use the boot-strapping technique suggested to derive p-values. See lines 240-247 as well as Tables 2 and 3.

The hypothesis is that the transmission intensity should be affected by both the source bat and the receiver. So one approach would be to construct the model to predict the infection intensity of every receiver bat (but not the source bat). In this model, the receiver’s infection intensity

(response) is a function of the receiver's personality, the source bat's personality, the sex composition of actor and receiver (MM, FF, FM), and the temperature. Trial is a random effect.

Response: Assuming we are interpreting this comment correctly, the reviewer is suggesting that we include all transmission and acquisition data in the same model. We explored this option and we are hesitant to present this model for two reasons: 1) it would be extremely large with many predictor variables and many interactions eating up a lot of degrees of freedom (see below, but there would be 8 interaction terms between all combinations of personality of transmitter/receiver and sex of transmitter/receiver); 2) the levels within the "transmission" data would be replicated within trials, which is okay in some circumstances, but is likely problematic with our sample size.

Specifically, the model would look like this:

```
Infection ~ activity_Y*sex +  
  activity_H*sex +  
  explore_H*sex +  
  sociability_Y*sex +  
  activity_Y_infected *sex_infected +  
  activity_H_infected *sex_infected +  
  explore_H_infected *sex_infected +  
  sociability_Y_infected *sex_infected +  
  TA + random=ID
```

Where "infected" is the personality and sex of the infected bat predicting the infection of the uninfected bats.

To avoid running such a large model with our dataset, as noted above, we opted to run a single transmission model and a single acquisition model. For the acquisition model, we used the permutation analysis suggested below. See tables 2 and 3 as well as Figure 2 for the coefficients from these models.

Personality there could be three independent PCs (like activity, exploration, and sociality) but not correlated measures (like activity 1 and activity 2). I would also present plots of each of these variables as sole predictors of infection intensity. They could also use AIC for model selection. Or perhaps personality could be just one variable that captures activity.

Response: As noted above (and suggested by the Associate Editor), we have opted to include both activity variables but we now also specifically note that the variance inflation factors for this model were low (<2.2, indicating limited multicollinearity in the dataset).

My suggestion (not a requirement just trying to be helpful):

You can use a permutation test to deal with the non-independence in your data. Rather than use degrees of freedom to get your p-value, create a null model to simulate the null hypothesis. For example, in your null model you can swap the infection intensity among the receivers within each trial to remove the relationship between receiver personality and infection density (while

keeping the total amount of infection the same within each trial).

Response: Thank you! This is helpful. As suggested, we now use this permutation test (and a slightly modified version of the code provided below) to ensure that animal personality and infection intensity were independent from one another in our null model. See lines 240-247 of the revised manuscript and Table 2.

Here's some r code as an example. Definitely check this over carefully, because I'm not testing it with any data. It's just to give you an idea of how to do it.

Response: Thank you for the suggestion and for the R code. We have used a slightly modified version of this code and found this extremely helpful. We've posted on our github site for review (<https://github.com/qwebber/uv-powder>). [note, we removed code from the response to reduce the length of the document].

Problem #2 The null hypothesis is already way too unlikely

I think this experiment is valuable but not for the reason described in the paper. Let's start with this question: Are more active animals more likely to spread and acquire a disease? Of course. Do we need a demonstration of that fact? Is there any chance that differences in movement rate has no causal influence on social transmission in the real world? I don't think so; the question is only the effect size and if it is large enough to matter given a set of ecological circumstances. The paper presents this as a hypothesis: that IF the personality trait of "activity" is correlated with movement and space use (which it is by definition), and IF these movement affects pathogen transmission (which of course it does), then this means personality COULD impact pathogen transmission...

Response: thank you for these thoughtful comments. We now present model estimates in Figure 3 as a means to assess the potential effect size. After significant work chasing down an approach in the literature, we have come to the conclusion that estimating effect sizes for GLMs and linear mixed models is not common and we don't know of a straightforward way to do this. We would be happy to present other effect size measures if the reviewer has any specific suggestions about resources we could look into.

But logically and mathematically, the effect of variation in movement rates on pathogen transmission cannot be zero. To the extent that personality predicts behavior at all (which is what it means to have personality), there must be an effect of being more active and more social on pathogen transmission (if transmission is by physical contact). I would say that this is an established fact. Every person should still believe this fact to be true regardless of what is reported in this paper.

So this is really an attempt at a demonstration. Fine. But when I see a table of insignificant effects, I just see a failure to detect an effect that we already know exists. The paper says "there was no effect of activity or sociability on infection transmission" but it is only talking about failing to detect the effect in the sample, not the effect in the population.

Response: see our comment above about effect sizes. We also now analyze our data using the permutation test suggested.

You might not detect the effect, but it essentially must be there. This is, by the way, how I feel about studies “discovering” personality in species X (or not). With the proper statistical power, one can detect personality in clonal fish raised in identical tanks since birth. I think there is no reason to test the null hypothesis anymore that individuals should be identical in their behavior. Similarly, the idea that host behavior affects pathogen transmission should be the null hypothesis.

The valid question is then not “Does variation in movement variation in pathogen transmission?” but rather questions like “How good are the correlations between these measures?” “How well can one predict an individual’s propensity to get a disease using standardized test of movement?” Or “If we capture some bats and measure their activity in these ‘personality tests’, how easy is it to infer their relative impact in a wild contact network?”

In other words, rather than trying to demonstrate, or infer the existence or absence of, these effects which we basically know to be true, it would be better to try to best measure the effect size estimates and their precision. These analyses could be used in meta-analyses and would be helpful for other people trying to measure the link between animal personality measures and outcomes like pathogen transmission using similar kinds of experiments. It could also help people with experimental design. How many bats do I need in my test? How variable is their behavior likely to be? In my mind, this is what the paper is about (or should be about). It’s not about “discovering” that host behavior is repeatable and affects pathogen dynamics. That’s a primary assumption of most or all disease ecologists, and I also don’t think the reader can infer that at least from the analysis they did.

Response: These are fair questions, to be sure and we appreciate these thoughtful ideas. We don’t necessarily agree with the reviewer’s ironclad confidence in the effects of personality on transmission. For example, just one scenario, what if colonial bats are so social that, even if there is variation in personality within the colony, there is effectively no influence on pathogen loads or prevalence because all bats in the colony end up interacting enough while day-roosting to result in rapid transmission to all colony members. Thus, we stand by our original “yes-no” hypothesis framework but we now include measures of effect size to address the reviewer’s important and thoughtful suggestions.

Problem # 3 Presentation

In my opinion, the scope of this paper and importance of the results is exaggerated way too much. For example, the first sentence reads,

“Given the current global pandemic of coronavirus disease 2019 (COVID-19), it is an understatement to say that infectious diseases of wildlife have become a major concern for human public health”

Is COVID-19 really the relevant place to start? The obvious place to start is with white-nose

syndrome, the important disease that is wiping out these bats.

Response: we have changed the beginning of the introduction to make it more general and remove examples (which we now include later where we introduce bats) but we strongly disagree that zoonotic disease and wildlife diseases with conservation impacts, are not both relevant starting points. Understanding more about how traits of individuals affect transmission of wildlife pathogens is not just important because of conservation implications. And particularly zoonotic pathogens of bats seem like an elephant in the room given the current global situation and hypothesized links between bats and SARS-CoV-2 and other high profile zoonotic pathogens. Moreover, in the case of our study species “these bats” (i.e., *Myotis lucifugus*) can host pathogens with clear public health implications (e.g., rabies virus) as well as pathogens that could become important for public health in the future (e.g., myotis coronavirus). On top of this, there is also reason to suspect that SARS-CoV-2 could spill back into North American bats from humans, creating a new wildlife reservoir. Thus, while we don’t want to belabour it, ignoring the zoonotic pathogen context seems like a glaring omission to us.

Several points concluded in the discussion cannot be inferred from the results in this paper:

The paper doesn’t present a test of an interaction effect of sex (i.e. that personality effect on infection differs by sex). Detecting an effect in only one sex is not the same thing as the slopes being different. The other inferences are not reliable due to problems described above.

Response: we now include sex in an interaction with each personality trait in our global model.

There are some confusing arguments made based on switching the meaning of social and sociability. The text needs to use these terms consistently.

Response: sociability is defined at lines 59-60 in the methods. We checked through the discussion and made sure our use of ‘sociability’ and ‘social’ was appropriate and consistent. We were unable to determine exactly where the reviewer was having issue with these terms.

It says that the study would be “impossible with free-ranging bats” which is not true. Here’s one example: Bakker et al. 2019. Fluorescent biomarkers demonstrate prospects for spreadable vaccines to control disease transmission in wild bats. Nature Ecology and Evolution.

Response: what we meant here was that our experimental set up enabled us to measure personality traits and recapture the same individuals to quantify transmission of the UV powder. At least for little brown bats at swarming sites it is nearly impossible to recapture the same individuals given the large number of bats present (>10,000 bats hibernate at the hibernacula where we conducted this study). However, have reworded to say “very difficult” instead of impossible and we now cite Bakker et al. 2019 at line 365.

There are also some excellent point discussed, such as, that more active/social bats might harbor more parasites, that pathogens can shape personality traits, and that temperature can lead to huddling and change contact networks.

Response: thank you for this comment.

The conclusion statement really oversells this study, and I think the text comes off as grandiose and hyperbolic which really detracts from the whole paper. I would completely delete or rewrite the conclusion. The text says: “Our results are among the first to experimentally manipulate environmental conditions such that the role of host behaviour could be identified as a potential mechanism regulating host-pathogen dynamics.” Putting aside the issues with the analysis, why would host behavior not have potential to regulate host-pathogen dynamics? What environmental conditions did they manipulate? It also says: “We also provide a starting point for future work to inform predictions about how host behaviour affects pathogen transmission in natural systems.” (??) In my view, the effect of host behavior on pathogen transmission is a large field. For example, look at recent reviews and papers by Matthew Silk, Daniel Streicker, Sonia Altizer, then there’s all the works on insects as vectors, etc. That’s all before you get to human epidemiology! The classic paper on superspreaders is 15 years old now.

Response: we have altered the conclusion and we now cite Silk et al. 2019 *Philos. Transactions R. Soc* at line 401 and Bakker et al. 2019 *Nature Eco Evo* at line 365. We now clarify how we manipulate “the environment” and make the conclusions and suggestions for future work a little more focused on bats to avoid over-generalizing.

The paper says it “clearly highlight the role of demography and environmental conditions” in pathogen dynamics (??). How? The experiment is with 10 captured groups of bats in a tent, where one has UV dust. Having two sexes becomes “demography”, having a temperature is the “environment”, and the powder is the “pathogen”. But there are many studies that study actual demography in natural environments with real pathogens, so it seems strange to bring up these factors here.

Response: we have re-written the conclusion and we now mention that our results highlight the potential role of sex-specific differences in behaviour could also influence pathogen dynamics. See lines 392-402.

When people study social network impacts on disease transmission (like effects of centrality) in a population that is socially structured (rather than spatially structured), it is assumed that these network measures are variable across individuals and repeatable within individuals, so this is the same thing as saying “personality” in different words. That’s one of the reasons this idea is not particularly novel in my opinion; it is already assumed by social network models of disease transmission.

Response: this is a good point, but we argue that animal personality is different than social network characteristics. Social networks are inherently complex and in most cases rely on two or more animals sharing space at the same time – which can definitely be a driver of pathogen transmission. Using centrality or degree (i.e. number of connections) as a predictor of pathogen load is great if you are interested in the role of social behaviour, but so many factors go into an individual having high centrality or degree; whereas we argue that personality is a more precise measure of individual behavioural variation and in fact represents a different set of

questions/expectations. While the role of personality as a driver of pathogen dynamics may not be especially novel anymore, it is still different than social networks.

Basically, the tone and scope of the discussion needs to be changed to match the scope of the results. This is not a paper that should change anyone's fundamental understanding of how host behavior might impact pathogen dynamics. It's not an impressive study in disease ecology. But that said, I think it is a neat experiment and could be a useful paper.

Response: we have changed the tone of the discussion. See lines 315-320 and 392-402 for some changes.

Although not a completely novel idea, I so like the idea of discussing behavior and individual variation in behavior as a factor in pathogen transmission, and I think there is much good work to do on that topic (which is one of the reasons this study provides useful data).

Response: we have slightly expanded our discussion of individual behavioural variation as a driver of pathogen transmission at lines 315-320.

Finally, it would be better to present all the relevant scatterplot correlations (in the supplement), not only the ones that were significant.

Response: we have added scatterplot figures with associated correlation coefficients for all combinations of variables as supplementary Figure S3.

Summary

I'm sorry to be so negative. If my tone seemed disrespectful, rude or unkind, please know that was not my intention. Despite the problems as written now, this paper has useful information in it that I might even use in my own research. For example, the data here would help me do a power analysis when designing similar experiments on different questions. I hope my extensive criticism does not discourage the authors and that my suggested revisions improve the paper.

Response: thank you again for the detailed comments on the manuscript.

Other minor line by line comments

Regarding the title, I would discourage the use of the phrase "proxy pathogen". To me that sounds like a tractable or safer pathogen used to study another less tractable pathogen. Also the use of transmission and acquisition in the title is redundant because you cannot increase just one of them without increasing the other.

Response: we have changed the title to "personality affects dynamics of an experimental pathogen in little brown bats".

I would just say "Evidence that personality affects pathogen transmission in little brown bats" or something similar (if they indeed can find that).

Abstract (and also maybe the methods) needs to explain experimental design better. Range of bats in each group? How many groups? How many trials? Are any bats used in multiple groups?

Response: we have revised the abstract to address these questions. See lines 26-29.

Line 61. Is personality a behavioral “mechanism”? Not in my mind. I would call personality a behavioral phenotype or a factor that affects transmission. It describes proclivity towards behaviors that determine how often or how quickly a pathogen could spread, but it does not explain how transmission works. For example, compassion is not a “mechanism” of violence, but it is a factor that might influence whether violence occurs or not.

Response: we have changed “mechanism” to “phenotype”. See line 57.

Line 68. *Many* bats are highly gregarious (not all)

Response: fixed as suggested.

Rephrase for clarity: “social structure appears ephemeral”

Response: we have reworded as “during swarming, social associations tend to be ephemeral and non-preferential”. See line 73-74.

Line 93. You should say here that you captured and tested bats that were swarming to link to the above text.

Response: we now mention that bats were captured at swarming sites. See lines 94 and 110.

Line 97. What does “naïve” mean here? If it means “uninfected”, then maybe just say that.

Response: we have changed “naïve” to “uninfected” throughout the manuscript.

Be clear that your experiment is not measuring who gets what from whom. A bat could acquire the pathogen directly from the experimentally infected bat or indirectly from another bat or the enclosure wall, right? So I would say that the second hypothesis is that the more active bats are more likely to acquire the pathogen (but not specify how or from whom).

Response: we have removed the aspect of this hypothesis that suggested they would only acquire the pathogen from the originally infected bat. See line 104.

Line 149. Why unique? There are many studies on how variation among individuals affects infection dynamics.

Response: we have removed this sentence.

Line 182. An important aspect of measuring personality is putting animals in a standard

environment. In one trial, the stimulus bat might call with echolocation or even social calls and in another trial it might be silent. The behavior of the stimulus bat might matter more than the personality of the focal bat, so this is likely to be a noisy measure of sociality (and might be worth mentioning this as a caveat).

Response: we have added this as a caveat. We also cite Webber and Willis (2020) Behaviour as another example of where we use this test and outline some of the caveats associated with this test. See lines 194-195.

I think Table S2 should be in the main text (if the journal allows this amount of space).

Response: we have moved this table (ethogram definition of behaviours) to the main text – see Table 1.

Line 240 Code is available...

Response: fixed.

Line 293 typo : ‘that’

Response: fixed.

369 “The negative binomial distribution of infection intensity we observed was almost identical to the pattern observed for natural parasites of free-ranging bats” If this is true (almost identical), then this is maybe the most interesting part of the paper in my opinion. Please present the model fit and an analysis!

Response: we have moved this figure to the main text (new Figure 1) and also conducted model fit for the negative binomial distributions. See lines 259-261.

Reviewer: 2

Comments to the Author(s)

This is a very nicely designed and carried out study, able to explore the causality of personality in explaining variation in pathogen spread. The paper is clearly written, analyses sound and results soundly discussed. I only have 2 v minor comments:

Response: thank you for your comments.

L66: I suggest that this reference is of relevance here:
Petkova et al (2018). Parasite infection and host personality: Glugea infected three-spined sticklebacks are more social. BEHAVIORAL ECOLOGY AND SOCIOBIOLOGY 72:173.
doi:10.1007/s00265-018-2586-3.

Response: we now cite Petkova et al. (2018) at line 62.

L151-: it is not clear that only 1 individual per group was infected by the text here, neither how this was chosen (in terms of personality). Also, was personality assessed prior or after observations? Just double check that these aspects are clear.

Response: we have re-arranged some of the sentences here and moved the original first sentence from this section to note that we randomly selected a single bat per group to be the 'infected' individual. See lines 150-151.

Appendix B

Associate Editor Comments to Author (Dr Kimberley Mathot):

Comments to the Author:

Thank you for submitting the revised version of your manuscript. I'm satisfied that you have addressed the major comments/concerns raised in the initial review, but have a number of very small edits I would like you to make. Several of these relate to your use of the term "personality" to refer to individuals. Personality is a population level characteristic (a measure of the proportion of phenotypic variance in a population that exists at the among-individual level) - not an individual attribute. As such, statements like "the personality of the infected individual" are technically incorrect.

Response: thank you for these comments and for highlighting the mis-use of the term personality – we appreciate it and have made all of the recommended changes.

Minor edits

Line 18: please rephrase to “We quantified activity and sociability in little brown bats (*Myotis lucifugus*) and then experimentally....”

Lines 20-21: please rephrase to : “1) more sociable and more exploratory individuals...”

Line 22: please rephrase to: “...more sociable and exploratory....”

Lines 29-31: please rephrase to “As predicted, the exploratory behaviour of the experimentally infected individual was positively correlated with infection intensity in their group-mates, while more”

Lines 56: please rephrase to “Consistent among-individual....”

Line 89: replace “within” with “during”.

Line 96: replace “personality” with “behavioural”

Line 97: replace “personality” with “behavioural type”

Line 101: replace “personality” with “behavioural type”

Line 173: rephrase to “....., all personality traits measured here have previously been shown to be weakly to highly repeatable...”

Line 227: replace “personality traits” with “behavioural traits”

Line 292: replace “personality composition” with “behavioural composition” and replace “host personality” with “host behaviour”

Line 348-349: rephrase to “Our finding could be important for understanding pathogen transmission dynamics from both conservation and public health perspectives”.

Line 394-395: rephrase to “..... we were able to identify behavioural and environmental...”

Line 398: rephrase to “host behaviour”

Line 636: rephrase to “the effect of behavioural of a single...”

Response: we have made all of these changes as suggested in the revised manuscript.

It's of course a personal choice, but I would also recommend acknowledging the referee that signed their review and provided you with sample code.

Response: we have now acknowledged the reviewers – however, after looking closely at the original files it doesn't seem like this reviewer signed their review (at least as far as I could tell).